

**Formation and growth of atmospheric nanoparticles in the eastern Mediterranean: Results**
**from long-term measurements and process simulations**
Nikos Kalivitis[1,2], Veli-Matti Kerminen[3], Giorgos Kouvarakis[1], Iasonas Stavroulas[1,4], Evaggelia
Tzitzikalaki[1], Panayiotis Kalkavouras [1,5], Nikolaos Daskalakis[1,6], Stelios Myriokefalitakis[5,7],
Aikaterini Bougiatioti[5], Hanna Elina Manninen[8], Pontus Roldin[9], Tuukka Petäjä[3], Michael Boy[3],
Markku Kulmala[3], Maria Kanakidou[1], and Nikolaos Mihalopoulos[1,5]
1. Environmental Chemical Processes Laboratory, Chemistry Department, University of Crete,
70013, Heraklion, Greece
2. Institute for Astronomy, Astrophysics, Space Applications and Remote Sensing, National
Observatory of Athens, I. Metaxa & Vas. Pavlou, 15236 Palea Penteli, Greece
3. Institute for Atmospheric and Earth System Research, Gustaf Hällströmin katu 2, P.O. Box
64, FI-00014 University of Helsinki
4. Energy, Environment and Water Research Center, The Cyprus Institute, Nicosia 2121,
Cyprus
5. Institute for Environmental Research & Sustainable Development, National Observatory of
Athens, I. Metaxa & Vas. Pavlou, 15236 Palea Penteli, Greece
6. Laboratory for Modeling and Observation of the Earth System (LAMOS), Institute of
Environmental Physics (IUP), University of Bremen, Bremen, Germany,
7. Institute for Marine and Atmospheric Research (IMAU): Utrecht, Netherlands
8. Experimental Physics Department, CERN, 1211 Geneva, Switzerland
9. Lund University
*Correspondence to*: Nikos Kalivitis (nkalivitis@uoc.gr)



Abstract
Atmospheric New Particle Formation (NPF) is a common phenomenon all over the world. In
this study we present the longest time series of NPF records in the eastern Mediterranean
region by analyzing seven years of aerosol number size distribution data obtained with a
mobility particle sizer. The measurements were performed at the Finokalia environmental
research station on Crete, Greece during the period June 2008-June 2015. We found that NPF
took place 29% of the available days, undefined days were 26% and non-event days 45%. NPF
is more frequent in April and May probably due to the biogenic activity and is less frequent in
August and November. The NPF frequency increased during the measurement period, while
particle growth rates showed a decreasing trend, indicating possible changes in the ambient
sulfur dioxide concentrations in the area. Throughout the period under study, we frequently
observed production of particles in the nucleation mode during night-time, a feature rarely
observed in the ambient atmosphere. Nucleation mode particles had the highest
concentration in winter, mainly because of the minimum sinks, and their average contribution
to the total particle number concentration was 9%. Nucleation mode particle concentrations
were low outside periods of active NPF and growth, so there are hardly any other local sources
of sub-25 nm particles. Additional atmospheric ion size distribution data simultaneously
collected for more than two years period were also analyzed. Classification of NPF events
based on ion measurements differed from the corresponding classification based on mobility
spectrometer measurements, possibly indicating a different representation of local and
regional NPF events between these two measurement data sets. We used MALTE-box model
for a simulation case study of NPF in the eastern Mediterranean region. Monoterpenes
contributing to NPF can explain a large fraction of the observed NPF events according to our
model simulations. However the parametrization that resulted after sensitivity tests was
significantly different from the one applied for the boreal environment.
1) Introduction
Most of the atmospheric aerosol particles, and a substantial fraction of particles able to act as
cloud condensation nuclei (CCN), have been estimated to originate from new particle
formation (NPF) taking place in the atmosphere (Spracklen et al. 2006; Kerminen et al., 2012;
Gordon et al., 2017). The exact mechanisms driving atmospheric NPF and subsequent particle
growth processes are still not fully understood, nor are the roles of different vapors and ions
in these processes (Kulmala et al., 2014; Lehtipalo et al., 2016; Tröstl et al., 2016). In order to
understand how aerosol particles affect regional and global climate and air quality, it is



necessary to quantify the factors that determine the occurrence of NPF and characterize the
parameters that describe the strength of NPF, such as the new particle formation and growth
rates, in various environments.
While NPF has been reported to take place worldwide (Kulmala et al., 2004a; Wang et al.,
2017), observational studies on this subject are scarce in rural sub-tropical environments.
Several studies have investigated NPF in eastern Mediterranean and found it to be a frequent
phenomenon. Lazaridis et al. (2006) first reported NPF at the area and correlated these events
to polluted air masses. Petäjä et al. (2007) presented NPF in Athens metropolitan area and
showed that under the influence of urban pollution, condensing species leading to growth of
the new particles are far more hygroscopic than under cleaner conditions. NPF events have
also been reported to be frequent at the urban environment of Thessaloniki (Siakavaras et al.,
2016). Kalivitis et al. (2008) showed that precursors and nucleation mode particles experience
strong scavenging on Crete island during summer. Pikridas at al. (2012) suggested that
nucleation events occurred only when particles were neutral, being consistent with the
hypothesis that a lack of $NH_3$, during periods when particles are acidic, may limit nucleation in
sulfate-rich environments such as the eastern Mediterranean. Additionally, based on ion
observations, Pikridas et al. (2012) showed that NPF is more frequent in winter. By using the
same data set from eastern Mediterranean, Kalivitis et al. (2012) reported night-time
enhancements in ion concentrations with a plausible association with NPF, being among the
very few locations where such observations have been made. Manninen et al. (2010)
presented an analysis of a full year of observations of NPF with atmospheric ion spectrometers
at various locations across Europe during the EUCAARI project and showed that NPF is less
frequent in the eastern Mediterranean site than in other, mostly continental, European sites.
On the other hand, Berland et al. (2017) showed that similar patterns are being observed
throughout the Mediterranean when comparing observations from the island of Crete to a
western Mediterranean site in terms of the frequency of occurrence, seasonality, and particle
formation and growth rates. Kalivitis et al. (2015) studied for the first time the NPF-CCN link
using observations of particle number size distributions, CCN and high-resolution aerosol
chemical composition for the eastern Mediterranean atmosphere. From the hygroscopicity of
the particles in different size fractions, it was concluded that smaller particles during active
NPF periods tend to be less hygroscopic (and richer in organics) than larger ones. Finally,
Kalkavouras et al. (2017) reported that NPF may result in higher CCN number concentrations,
but the effect on cloud droplet number is limited by the prevailing meteorology.



In this work, we present results from the analysis of seven years of aerosol particles number
size distributions and more than two years of atmospheric ion size distributions, representing
the longest published NPF data set in the Mediterranean atmosphere. The main questions we
wanted to address were: 1) How often does NPF take place in eastern Mediterranean, what
are the characteristics of this phenomenon and to what extent has it changed over the period
under study? 2) Are there features in NPF observed at the study area that are not common in
other locations? and 3) How well can numerical models, used in different environmental
conditions, represent NPF in this subtropical environment?
2) Materials and methods
2.1) Measurements
Measurements presented in this work were carried out at the atmospheric observation
station of the University of Crete at Finokalia, Crete, Greece (35˚20′N, 25˚40′E, 250m a.s.l)
over seven years, between June 2008 and June 2015. The Finokalia station
(http://finokalia.chemistry.uoc.gr/) is a European supersite for aerosol research, part of the
ACTRIS (Aerosols, Clouds, and Trace gases Research Infrastructure) Network. The station is
located at the top of a hill over the coastline, in the north east part of the island of Crete
(Mihalopoulos et al., 1997). The station is representative for the marine background
conditions of eastern Mediterranean (Lelieveld et al., 2002), with negligible influence by local
anthropogenic sources. The nearest major urban center in the area is Heraklion with
approximately 200 000 inhabitants, located about 50 km to the west of the station.
In order to monitor the NPF events from the early stages of nucleation, a TROPOS type
custom-built Scanning Mobility Particle Sizer (SMPS), similar to IFT-SMPS in Wiedensohler et
al. (2012), was used at Finokalia. Particle number size distributions were measured in the
diameter range of 9–848 nm every five minutes. The system was a closed-loop, with a 5:1
ratio between the aerosol and sheath flow and it consists of a Kr-85 aerosol neutralizer (TSI
3077), a Hauke medium Differential Mobility Analyzer (DMA) and a TSI-3772 Condensation
Particle Counter (CPC). The sampling was made through a $PM_{10}$ sampling head and the sample
humidity was regulated below the relative humidity of 40% with the use of Nafion® dryers in
both the aerosol and sheath flow. The measured number size distributions were corrected for
particle losses by diffusion on the various parts of the SMPS according to the methodology
described in Wiedensohler et al. (2012). Three different types of calibration were performed
for the SMPS, DMA voltage supply calibration, aerosol and sheath flows calibrations and size



calibrations. These measurements have been performed at Finokalia on a continuous basis
since 2008. The instrument used at Finokalia was audited on-site with good results in the
framework of EUSAAR (European Supersites for Atmospheric Aerosol Research) project
(http://www.wmo-gaw-wcc-aerosol-physics.org/audits.html) and has successfully passed
twice laboratory intercomparison workshops (2013 and 2016, reports available at
http://www.wmo-gaw-wcc-aerosol-physics.org/instrumental-workshops.html) in the
framework of ACTRIS project. The instrument has been operated following the
recommendations described in Wiedensohler et al. (2012). Additional information for newly
formed particles were obtained with the use of an Air Ion Spectrometer (AIS- AIREL Ltd.,
Institute of Environmental Physics, University of Tartu, Estonia). AIS is a cluster ion air
spectrometer used to simultaneously measure electrical mobility distribution of positive and
negative air ions (mobilities in the range of 2.4 to $1.3 \cdot 10^{-3}$ cm$^2$ V$^{-1}$ s$^{-1}$). The mobility distributions
were then transformed to size distributions in the size range 0.8-42 nm. The number counting
threshold was approximately 10 cm$^{-3}$ and the uncertainties of the AIS measurements were
~10% for negative and positive ion concentrations and ~0.5 nm in size. The diameter of the
AIS inlet tube was 35 mm and the sample flow rate was 60 L m$^{-1}$. The time step of the
measurements was five minutes.
These measurements have been used to identify NPF for the whole period and provide a
historical perspective for the frequency and the characteristics of NPF phenomena in the
eastern Mediterranean. Calculations for formation rates of new particles (J), growth rates (GR)
in various size ranges and condensation sink (CS) were made according to Kulmala et al.
(2012). Formation rates of particles with diameter D were calculated as:

$$J_{Dp} = \frac{\Delta N_{D_p}}{\Delta t} + CoagS_{D_p} \cdot N_{D_p} + \frac{GR}{\Delta D_p} \cdot N_{D_p} + S_{losses} \ (1)$$

$\Delta N_{D_p}$ is the increase in nucleation mode particles' number concentration (D$_p$<25nm), CoagS
is the coagulation of particles in this size range, GR is the growth rate in the size range 9-25nm.
S$_{losses}$ takes into account additional losses and was neglected in this study. GR was calculated
using the mode-fitting method. The aerosol size distributions were fitted with lognormal
distributions and the nucleation mode geometric mean diameter was plotted as a function of
time. GR was calculated as the slope of the linear fit so that:

$$GR = \frac{dD_p}{dt} \ (2)$$

CS is the sulfuric acid sink caused by the preexisting aerosol population with unit s$^{-1}$.





All important meteorological parameters were monitored every five minutes using an
automated meteorological station, including the temperature, wind velocity and direction,
relative humidity, solar irradiance and precipitation. Ozone concentrations were measured
with a TEI 49C instrument and nitrogen oxides with a TEI 42CTL, both commercially available,
with a time step of five minutes.
2.2) NPF simulations with the MALTE-Box model
The simulations of NPF events in the eastern Mediterranean atmosphere were here
performed using the MALTE-box model of the University of Helsinki. This 0-d model able to
simulate aerosol dynamics and chemical processes has successfully reproduced observations
of aerosol formation and growth in the boreal environment (Boy et al., 2006) as well as in
highly polluted areas (Huang et al., 2016). For the present study, relevant chemical reactions
from the Master Chemical Mechanism were incorporated in the MALTE-box chemical
mechanism, as described in Boy et al. (2013). These include the full MCM degradation scheme
of the following volatitle organic compounds (described in more detail in Tzitzikalaki et al.,
2017): $C_1$-$C_4$ alkanes, $C_2$-$C_3$ alkenes, acetylene, isoprene, α- and β-pinene, aromatics,
methanol, dimethyl sulfide, formaldehyde, formic and acetic acids, acetaldehyde,
glycoaldehyde, glyoxal, methylglyoxal, acetone, hydroxyacetone, butanone and marine
amines. The Kinetic PreProcessor (KPP) was used to produce the Fortran code for the
calculations of the concentrations of each individual compound (Damian et al., 2002), except
for those species whose concentrations were manually input from large scale model
simulations.
The major aerosol dynamical processes for clear sky atmosphere were simulated by the size-
segregated aerosol model UHMA (University Helsinki Multicomponent Aerosol Model,
Korhonen et al., 2004) impended in the MALTE-Box model. Measured aerosol number size
distributions were used to initialize UHMA daily, which simulates NPF, coagulation, growth
and dry deposition of particles. UHMA simulated new cluster formation resulting from free
form nucleation. Apart from sulfuric acid, about 20 low-volatility organic compounds (ELVOCs)
and 7 selected semi-volatile organic compounds (SVOCs) were treated as condensing vapours,
following the simplified chemical mechanism presented in Huang et al. (2016). All these
compounds were treated as sulfuric acid and organics and the condensation of organic vapors
was determined by the nano-Kohler theory (Kulmala et al., 2004b).



As input to the MALTE-Box model were used the observations at Finokalia station and when
such observations were not available, the results from numerical simulations with the global
3-dimensional chemistry transport model (CTM) TM4-ECPL (Daskalakis et al., 2015, 2016;
Myriokefalitakis et al, 2010, 2016) for Finokalia. Observational data include temperature,
relative humidity, total radiation (meteorological input), ozone ($O_3$) and nitrogen oxides (NOx)
concentrations as well as aerosol number size distributions. The aerosol number size
distribution measured by the SMPS was used to calculate the condensation sink for $H_2SO_4$
vapors. Due to the lack of detailed measurements of VOC at Finokalia, as a first approximation,
biogenic and anthropogenic concentrations of all the above mentioned VOCs resolved every
3 hours were taken from the TM4-ECPL model.
The global TM4-ECPL model was run driven for this study by ECMWF (European Centre for
Medium – Range Weather Forecasts) Interim re–analysis project (ERA – Interim) meteorology
(Dee et al., 2011) of the year 2012 at an horizontal resolution of 3° in longitude x 2° in latitude
with 34 vertical layers up to 0.1 hPa. The model used year-specific meteorology and emissions
of trace gases and aerosols. For this study, that of the year 2012 was used, except for soil NOx
and oceanic CO and VOCs emissions which were taken from POET inventory database for the
year 2000 (Granier et al., 2005). TM4-ECPL simulations for this work were performed with a
model time-step of 30 min, and the simulated VOC concentrations every 3-hours were used
as input to MALTE box model; while $SO_2$ surface levels at Finokalia were taken from Monitoring
Atmospheric Composition and Climate (MACC) data assimilation system (Inness et al., 2013).
For the calculations of the photo-dissociation rate coefficient by the MALTE-Box model, the
solar actinic flux (AF) is needed. Unfortunately, AF was not measured at Finokalia in 2012,
therefore AF levels were calculated by the Tropospheric Ultraviolet and visible Radiation
Model (TUV, Madronich, 1993) version v.5 for cloud free conditions. The ability of TUV to
calculate the AF at Finokalia was investigated by comparing observations of photo dissociation
rates of $O_3$ ($JO^1D$) and $NO_2$ ($JNO_2$) and model calculations. The measurements of these photo
dissociation rates were performed by filter radiometers (Meteorologie Consult, Germany).
The $JO^1D$ was measured at wavelengths <325nm, while for $JNO_2$ wavelengths <420nm were
used.
A series of sensitivity tests of AF to different input parameters was also performed to optimize
the calculations. The model uses extra-terrestrial solar spectral irradiance (200-1000 nm by
0.01nm steps) and computes its propagation through the atmosphere taking into account
multiple scattering and the absorption and scattering due to gases and particles. TUV inputs



of interest were surface reflectivity (albedo), $O_3$ column, Aerosol Optical Depth at 500nm
(AOD), Single Scattering Albedo of aerosol (SSA), $NO_2$ column, air density. Total $O_3$ column
values were taken from Ozone Monitoring Instrument (OMI) on the Aura spacecraft of NASA
(Levelt et al., 2006). Aerosol columnar optical properties were obtained from the Aerosol
Robotic Network (AERONET). AOD data were measured at the FORTH_Crete station which is
located 35 km west of Finokalia (Fotiadi et al., 2006). Data level 1.5 was used (cloud-screened).
Total $NO_2$ column values were taken from GOME2 and OMI satellites. The calculations were
carried out at wavelength from 280 to 650nm with a resolution of 5nm. Simulations using
surface reflectivity of 0.075 and simulation using $O_3$ column taken from OMI had the best
correlation with measurements. However, the TUV model still significantly overestimated
$JO^1D$ and $JNO_2$ data. Thus, a parameterisation took place following a simple empirical
approach, according to Mogensen et al. (2015) and the ratios between the measured and
modelled (from TUV) photolysis rate were calculated.
3) Results and discussion
3.1) Particle size distribution and its connection with NPF
We analyzed all available measurements of number size distributions of atmospheric aerosol
particles measured at Finokalia in order to identify and analyze the NPF phenomenon in the
eastern Mediterranean. The data coverage for the period 2008-2015 was 82 %, providing the
longest time series of size distributions not only in this region but also in the southern Europe
and a unique data base for aerosol physical properties.
First, we calculated the total particle number concentration (median concentration was 2138
$cm^{-3}$) and corresponding number concentration in the nucleation mode ($d_p<25nm$, median 78
$cm^{-3}$), Aitken mode ($25nm<d_p<100nm$, median 992 $cm^{-3}$) and accumulation mode ($d_p>100nm$,
median 878 $cm^{-3}$). We found that Aitken mode accounted for 46% and accumulation mode
41% of the total particle number concentration, while the nucleation mode accounted only
for 3%. The standard deviation of the nucleation particle number concentration was 537 $cm^{-3}$,
indicating that the abundance of these smallest particles is of episodic nature. The highest
monthly average concentrations of nucleation mode particles were observed during winter
and the lowest ones during summer (Figure 1a). Calculating the median diurnal variability of
the nucleation mode, we can see that there is a clear pattern for all seasons of the year (Figure
2a) with a sudden burst in the number concentration around noon that is most pronounced
in winter and least in summer. Such an observation suggests that the nucleation particle



number concentration is controlled by NPF episodes. As can be seen in Figure 2b where a typical "banana shaped" pattern of an NPF event at Finokalia is presented, the sudden burst at noon is typical for a NPF event. In summer, nucleation mode particles have the highest concentrations during the night, yet another concentration relative maximum before noon can be attributed to NPF (Figure 2a). The shift in the average time of the daytime burst of nucleation mode particles can be attributed to the annual variation of the daylight length. Similar observations to ours have been reported in Cusack et al. (2013) for the western Mediterranean where the diurnal variation of nucleation mode particles presents a clear maximum at noon under both polluted and clean conditions.

It is worth noticing that during night-time the median nucleation mode particle number concentrations were almost the same in all the seasons. This suggests that there is some new particle production mechanism at night that operates separately from daytime NPF. Frequently during the night-time, we observed a pronounced appearance of new nucleation mode particles over several hours as illustrated by Figure 3. While nocturnal NPF has been reported in the literature (see Salimi et al. (2017) and references therein), this phenomenon seems to be rare and it remains unclear what are the exact mechanisms leading to it. Given that we observed no or little growth during nighttime NPF, we may assume that the sources leading to the formation of new particles are local rather than regional. Observations of very localized NPF have been reported in Mace Head, Ireland, where intense NPF frequently takes place under low tide conditions when algae are exposed to the atmosphere (O'Dowd et al., 2002). Henceforth, we will exclude the nighttime NPF events from our further analysis. We refer the interested reader to Kalivitis et al. (2012) for a more detailed description of this phenomenon.

Overall, we observed atmospheric NPF to take place during both day and night at Finokalia, but no sign of any other source of nucleation mode particles in measured air masses. We therefore hypothesize that atmospheric NPF is the dominant source of nucleation mode particles in this Mediterranean environment.

3.2) Characteristics of NPF in the eastern Mediterranean

We analyzed the dataset of aerosol size distributions following the approach of Dal Maso et al. (2005) in order to mark the available days as 1) NPF event days when a clear new nucleation mode and subsequent growth of newly-formed particles to larger diameters can be observed, 2) non-event days and 3) undefined days when either new particles appear into the Aitken


mode or nucleation mode particles do not show a clear growth. The available days were
manually inspected and classified.
We used the Statistica software package for Windows to carry out factor analyses, including
meteorological parameters, ozone concentrations (as the major oxidant in the atmosphere)
and $PM_{10}$ mass concentration (as an index of particulate pollutant levels), in order to examine
whether any of these factors were associated with the formation of new particles,
represented by the nucleation mode number concentration. Furthermore, we divided our
data to night and day time periods in order to separate daytime NPF from that taking place
during nighttime. The only parameter that had some effect on the nucleation mode particle
number concentration was the wind velocity: when strong winds were prevailing at Finokalia,
it was more unlikely to observe nucleation particles. On the other hand, the lack of correlation
to any other parameter may indicate that the NPF is not sensitive to local meteorological
conditions or atmospheric chemical composition in this environment. Air mass back
trajectories calculated using the HYSPLIT model showed no major difference during NPF
events from air masses typical for the prevailing situation at Finokalia: air masses arriving at
Finokalia from the northeast were the most frequent during NPF events (27% against 24% of
all days), followed by northwestern air masses that were more frequent than the average
(21% against 17%) and northern directions (18% against 20%).
Next, we focused on determining the main characteristics of daytime NPF at Finokalia. Overall,
623 NPF events were identified. This is the longest time series of the NPF phenomenon
recorded in the Mediterranean atmosphere, providing a representative climatology of NPF
events in this region. NPF took place 29% of the 2121 available measurement days whereas
no event occurred on 45% of those days. It is worth noting that 26% of the days were
characterized as undefined, which means that while no clear NPF event could be observed,
there was some evidence of secondary particle formation although not at the immediate
vicinity of the station (Table 1). We found that NPF is most frequent in April and May, probably
due to the biogenic activity, and least frequent in August and November (Figure 4) probably
due to high wind speeds occurring these months from NE and S/SW directions respectively
(not shown). Nevertheless, NPF takes place throughout the year. One would expect NPF to be
most frequent in winter when the highest concentrations of nucleation particles are observed,
however this was not the case. A possible explanation for the high nucleation mode particle
number concentrations in winter could be that the probability of a newly formed particle to
survive is larger than in other times of the year. The survival probability of newly-formed





particles is closely related to the ratio GR/CS (Kerminen and Kulmala, 2002; Kulmala et al.,
2017). By looking at the seasonal variability of CS and GR (Figures 1b and 6b), the particle
survival probability seems to be the highest in winter.
As a next step, we classified the NPF events into Class I or Class II events depending on whether
the particle formation rate at 9 nm ($J_9$) and growth rates from 9 to 25 nm diameter ($GR_{9-25}$)
could be calculated with a good confidence. Overall, Class I events corresponded to 8% of the
available measuring days and 26% of the event days, and they were observed throughout the
year, providing enough data for a statistical analysis of particle formation and growth rates
during NPF events (Figure 5).
The average value of $J_9$ during the Class I NPF events in Finokalia was $1.1 \pm 1.6 \ cm^{-3} s^{-1}$ (median
$0.5 \ cm^{-3} s^{-1}$). This is well in the range of values reported for $J_{10}$ in other locations (Kulmala et
al., 2004a), higher though than $J_{16}$ reported by Berland et al. (2017) at the Finokalia site in
2013 ($0.26 \ cm^{-3} s^{-1}$), but substantially lower than the values found by Kopanakis et al. (2013)
in western Crete ($13.1 \pm 9.9 \ cm^{-3} s^{-1}$). The monthly variation of $J_9$ (Figure 6a) shows that the
highest formation rates were observed in November and March. The spring maximum in the
particle formation rate might be due to the enhanced biogenic activity and increasing
photochemical activity. The November maximum might appear as a result of the initiation of
the rain season at Crete and the subsequent rapid drop in CS, even though it is very difficult
to say which factors determine the monthly variability of $J_9$ at Finokalia. Seasonal averages of
$J_9$, $GR_{9-25}$ and CS are summarized in Table 2. Moreover, we found that $J_9$ and $N_{9-25}$ have a clear
linear relation (Figure 7), which supports our earlier hypothesis that at Finokalia the main
source of nucleation mode particles is their secondary formation in the atmosphere.
We calculated the average growth rate of the newly formed particles to be $5.1 \pm 3.9 \ nm \ hr^{-1}$
(median $4.1 \ nm \ hr^{-1}$). We found that $GR_{9-25}$ is highest in summer and lowest in winter and early
spring, probably in line with the seasonal cycle of photochemical activity and biogenic
emission patterns, producing condensable species that are driving the growth process (Figure
6b). The average duration of the NPF in summer seems to be shorter as shown in Figure 2a
and that may be explained by the higher growth rates observed.
3.3) NPF trends during the 2008-2015 period
By looking at the inter-annual evolution of the NPF monthly event frequency at Finokalia for
the 85 available months, we observe a slight increase of about 1.5 % per year (Figure 8a). This
trend is not statistically significant since p-value was found to be 0.07. This increase is a result





of a notable increase of Class II NPF events despite a simultaneous decrease of Class I events
from 2008 to 2015 (8b). During the measurement period under study, no trend in $J_9$ was
observed, but the winters 2008-9 and 2012-13 had clearly higher values of $J_9$ than the rest of
the time (Figure 8c).
When looking at the temporal variation of GR (Figure 8d), we observe a clear decreasing trend
of about 0.3 nm hr$^{-1}$ yr$^{-1}$. This trend can be considered statistically significant, the no-trend
hypothesis test returned a p-value of 0.03. In order to explain this trend, we need to
emphasize the regional characteristics of the observations at Finokalia, as this site is greatly
affected by long-range transported pollutants of marine, desert dust and polluted continental
origin (Lelieveld et al., 2002). Non-sea salt sulfate (nss-$SO_4^{2-}$) can be considered as an indicator
of regional pollution from anthropogenic activities ($SO_2$ emissions), and since the beginning of
the economic crisis in Europe, especially in Greece, we can observe a clear decline in its
concentration (Paraskevopoulou et al., 2015). We can therefore assume also a regional
decrease in $SO_2$ emissions, since a major part of $SO_2$ at Finokalia can be attributed to
transported pollution (Sciare et al., 2003). This would result in a decrease in the availability of
sulfuric acid, a major condensable species responsible for the particle growth (Bzdek et al.,

17   2012).

Hamed et al. (2010) studied the effect of the reduction in anthropogenic $SO_2$ emissions in
Germany between the years 1996-97 and 2003-06 as a result of the socio-economic changes
in East Germany after the reunification. They observed a notable decrease in the NPF event
frequency but an increase in the growth rate of nucleated particles. A decrease in the NPF
frequency due to the reduction of anthropogenic $SO_2$ emissions in eastern Lapland was
observed by Kyrö et al. (2014), and this decrease was most pronounced for the Class I NPF
events. Nieminen et al. (2014) analyzed the longest data set reported in literature from
Finland and found that, despite major decreases in ambient $SO_2$ concentrations observed all
over Europe as a result of overall air quality improvements, there was a slight upward trend
in the particle formation and growth rates. This feature was attributed partly to increased
biogenic emissions over the same period. Taken together, we conclude that the observed
decrease in the particle growth rate and frequency in the most pronounced NPF events in
Finokalia could as well be due to decreased $SO_2$ concentrations. The reasons for the overall
increase in the NPF frequency and little change in $J_9$ remain unclear, even though factors like
meteorological conditions and organic vapor abundance have probably played some role in
this respect.





3.4) Atmospheric ion observations related to new particle formation
At the Finokalia station, atmospheric ion observations relevant to new particle formation were
performed during two separate periods, 2008-2009 during the EUCAARI project (Manninen et
al., 2010) and 2012-2014 during the FRONT (Formation and growth of atmospheric
nanoparticles) project. Here we will focus only on FRONT data, since the EUCAARI dataset is
discussed in detail in Manninen et al. (2010 ) and Pikridas et al., (2012). A typical nucleation
event is presented in Fig. 9 as recorded by both the AIS and SMPS. AIS observations may
provide information about the initial stages of new particle formation as particles can be
observed emerging in the intermediate ion diameter range 1.6-7.4 nm. Intermediate ions
appear only under certain circumstances, such as during precipitation, at high wind speeds,
and when NPF is taking place (Hõrrak et al. 1998; Tammet et al., 2014; Leino et al., 2016; Chen
et al., 2017). In the following we will focus on NPF and use only the observations from the
negative polarity due to the better representation of NPF events in those data compared with
corresponding positive ions in our dataset (Figure 9).
We classified all of the available AIS measurement days into event, non-event and undefined
days, once again according to methods introduced by Dal Maso et al. (2005), and subsequently
compared the findings from AIS data to those from the SMPS data. Surprisingly, the two data
sets for the same time period gave quite different results in terms of the NPF event frequency:
in the AIS data the NPF event frequency peaked earlier during the year than in the SMPS data
(Figure 10). This feature was evident in both periods of AIS measurements and is probably due
to the different measurement characteristics of the AIS and SMPS instruments. For example,
it is possible that AIS data are more representative of local NPF events with limited particle
growth, and such events may not be seen in the SMPS data. On the other hand, the SMPS
measures neutral particles but has a much higher detection limit (9nm), so its data may be
more representative of regional NPF that takes place over distances of hundreds of kilometers
(Kalkavouras et al., 2017).
We calculated the growth rates at three different size ranges for the FRONT project similarly
to Manninen et al. (2010) and Pikridas et al (2012) for the EUCAARI project data. The particle
growth rates in the size ranges 1.5-3 nm, 3-7nm and 7-20 nm were 1.6 ±1.8 nm hr$^{-1}$, 5.4±4.9
nm hr$^{-1}$ and 9.1±9.5 nm hr$^{-1}$, respectively. These values are lower than those in Pikridas et al.
(2012) but comparable to those observed during the EUCAARI project for the first two size
ranges, and higher than those observed during the EUCAARI project for the last size range
(Manninen et al., 2010). Overall, we observed much faster growth of newly-formed charged



particles in the eastern Mediterranean atmosphere after their first growth steps beyond 3 nm
in diameter, reflecting probably the strong Kelvin effect at small particle sizes preventing
condensation and hence growth, and the abundance of precursors leading to nucleation and
condensing species contributing to each growth stage.
3.5) Simulations of NPF using the zero-dimensional model MALTE-box
In order to evaluate our understanding of the observed NPF events in the eastern
Mediterranean we chose to simulate two distinct cases of one week duration each, during
which NPF events have been observed (event week) or not (no event week). The selection was
done from the summer of the year 2012, when $JO^{1D}$ and $JNO_2$ photodissociation
measurements were also available at Finokalia. Two weeks in August 2012 were chosen,
28/08– 03/09 as event week and 09/08– 15/08 as non-event week. The "event week" was
described in detail by Kalivitis et al. (2015). Applying the MALTE-Box model the aerosol size
distribution and its evolution over the week has been simulated for these two cases.
During the ''event week'' the simulated formation of new particles successfully coincided with
the observations, as shown in Figures 11a and 11b. The NPF levels simulated using the
nucleation rates as parameterized for the boreal environment overestimated the
observations while the simulated growth of newly-formed particles was greatly
underestimated as shown in Tzitzikalaki et al. (2017). The most likely reason for this is the very
low concentration of monoterpenes, calculated by TM4-ECPL global model for the Finokalia
model grid box, on which the ELVOC and SVOC chemistry was built on. Indeed, the TM4-ECPL
model results for Finokalia were too low compared to monoterpenes observations in 2014
(not shown). Therefore, we performed a number of sensitivity tests to improve the
simulations. The best agreement between model results and observations was reached by
decreasing the nucleation coefficient from $10^{-11}\,s^{-1}$ (the value commonly used for the boreal
environment) to $5\times10^{-16}\,s^{-1}$ and increasing by a factor of 10 the α- and β-pinene
concentrations. With these modifications the model results greatly improved and the aerosol
number size distributions were well captured, as well as total number and volume
concentration of aerosol particles (Figures 11c and d respectively). This was the first time that
we were able to simulate in such detail NPF in the eastern Mediterranean. The almost five
orders of magnitude lower nucleation coefficient used here for the sub-tropical set-up could
be related to the contribution of still unknown compounds in the cluster-formation process.
Huang et al. (2016) applied different kinetic nucleation coefficients at Nanjing, China, with the
lowest value for a "China-clean" day of $6.0 \times 10^{-13}\,s^{-1}$.





Using the non-event week as our control case, we performed simulations of number size
distributions at Finokalia station using the sub-tropical set-up and compared it to our
measurements. For the "non-event week", weak NPF were predicted by the model during the
last two days that were not found in the measurements (Tzitzikalaki et al., 2017) but appear
to be associated with the rapid drop of CS during day five of the simulations. Nevertheless,
even if no NPF took place during the last two days, it was apparent in our measurements that
some nucleation particles appeared (~200 cm$^{-3}$) and thus the general tendency was captured
by the model. Both total number and volume concentrations were well captured by the model
(Figures 12 a, b). These results show the potential of MALTE-box model to simulate the NPF
in the eastern Mediterranean and the importance of input data. Therefore, when more
appropriate input data for Malte-box will become available (concurrent detailed
measurements of gases and aerosol distributions) at Finokalia, new detailed simulations will
further provide insight in NPF phenomena and the factors controlling them in the eastern
Mediterranean atmosphere.
4) Conclusions
NPF in the atmosphere is a recurrent phenomenon in eastern Mediterranean. In this study,
we presented the longest time series of NPF records in the region. We analyzed 2121 days of
aerosol number size distribution data from June 2008 until June 2015 and found that NPF took
place 29% of the available days, more frequently in spring and less frequently in late summer
and autumn. Production of nucleation mode particles was common during night-time as well.
Nucleation mode particle number concentrations were low outside periods of active NPF and
subsequent particle growth indicating absence of local sources. Classification of NPF events
based on atmospheric ion measurements differed from the corresponding classification based
on mobility spectrometer measurements: the maximum frequency of NPF events was
observed earlier in spring from AIS data than from SMPS data, possibly indicating a different
representation of local and regional NPF events between these two data sets since SMPS
measures new particles after they have grown to diameters larger than 9nm and hence
records only regional events lasting for several hours.
During the measurement period, the frequency of NPF occurrence increased by 1.5 % per year
while the average GR decreased by 0.3 nm hr$^{-1}$ yr$^{-1}$, probably reflecting the decrease of
ambient SO$_2$ concentrations due to the economic crisis. We used the MALTE-box model to



simulate NPF observations in the eastern Mediterranean region. Using a "sub-tropical"
environment parametrization, we were able to simulate with good agreement the selected
time period. The parametrization used was significantly different than the one used for the
boreal environment: nucleation rates were much lower, yet monoterpenes seemed to play a
key role in the mechanisms governing NPF phenomena.
From the results presented in this work it is evident that the Finokalia site is a unique location
in the eastern Mediterranean for studying the processes leading to NPF in the marine
environment. As a next step, a more detailed look to the precursors driving these processes is
necessary, with special emphasis on VOCs, and the expansion of the available measurements
at the site in order to eliminate the uncertainties introduced in our simulations from the use
of model outputs instead of observations.
5) Acknowledgements
The research project was implemented within the framework of the Action «Supporting
Postdoctoral Researchers» of the Operational Program "Education and Lifelong Learning"
(Action's Beneficiary: General Secretariat for Research and Technology), and was co-financed
by the European Social Fund (ESF) and the Greek State. This research is supported by the
Academy of Finland Center of Excellence program (project number 1118615). We
acknowledge funding from the EU FP7-ENV-2013 program "impact of Biogenic vs.
Anthropogenic emissions on Clouds and Climate: towards a Holistic UnderStanding"
(BACCHUS), project no. 603445 and the Horizon 2020 research and innovation programme
ACTRIS-2 Integrating Activities (grant agreement No. 654109). This study contributes to
ChArMEx work package 1 on aerosol sources.



6) References
Berland, K., Rose, C., Pey, J., Culot, A., Freney, E., Kalivitis, N., Kouvarakis, G., Cerro, J. C.,
Mallet, M., Sartelet, K., Beckmann, M., Bourriane, T., Roberts, G., Marchand, N.,
Mihalopoulos, N., and Sellegri, K.: Spatial extent of new particle formation events over the
Mediterranean Basin from multiple ground-based and airborne measurements, Atmos. Chem.
Phys., 17, 9567-9583, https://doi.org/10.5194/acp-17-9567-2017, 2017.
Boy, M., Hellmuth, O., Korhonen, H., Nillson, D., ReVelle, D., Turnipseed, A., Arnold, F. and
Kulmala, M.: MALTE – Model to predict new aerosol formation in the lower troposphere,
Atmos. Chem. Phys., 6, 4499–4517, doi:10.5194/acp-6-4499-2006, 2006.
Boy, M., Mogensen, D., Smolander, S., Zhou, L., Nieminen, T., Paasonen, P., Plass-Dülmer, C.,
Sipilä, M., Petäjä, T., Mauldin, L., Berresheim, H., and Kulmala, M.: Oxidation of $SO_2$ by
stabilized Criegee intermediate (sCI) radicals as a crucial source for atmospheric sulfuric acid
concentrations, Atmos. Chem. Phys., 13, 3865-3879, https://doi.org/10.5194/acp-13-3865-

14  2013, 2013.

Bzdek, B. R., Zordan, C. A., Pennington, M. R., Luther, G. W., and Johnston, M. V.: Quantitative
Assessment of the Sulfuric Acid Contribution to New Particle Growth. Environmental Science
& Technology, 46, 4365–4373. http://doi.org/10.1021/es204556c, 2012.
Chen, X., Virkkula, A., Kerminen, V.-M., Manninen, H. E., Busetto, M., Lanconelli, C., Lupi, A.,
Vitale, V., Del Guasta, M., Grigioni, P., Väänänen, R., Duplissy, E.-M., Petäjä, T., and Kulmala,
M.: Features of air ions measured by an air ion spectrometer (AIS) at Dome C, Atmos. Chem.
Phys., 17, 13783-13800, 2017.
Cusack, M., Pérez, N., Pey, J., Wiedensohler, A., Alastuey, A., and Querol, X.: Variability of sub-
micrometer particle number size distributions and concentrations in the Western
Mediterranean regional background. Tellus B, 65.
doi:http://dx.doi.org/10.3402/tellusb.v65i0.19243, 2013.
Dal Maso, M., Kulmala, M., Riipinen, I., Wagner, R., Hussein, T., Aalto, P. P., and Lehtinen, K.
E. J.: Formation and growth of fresh atmospheric aerosols: eight years of aerosol size
distribution data from SMEAR II, Hyytiälä, Finland, Boreal Environ. Res., 10, 323–336, 2005.



Damian, V., Sandu, A., Damian, M., Potra, F., and Carmichael, G. R.: The kinetic preprocessor
KPP-a software environment for solving chemical kinetics, Computers & Chemical
Engineering, 26, 1567–1579, http://doi.org/10.1016/S0098-1354(02)00128-X, 2002.
Daskalakis, N., Myriokefalitakis, S., and Kanakidou, M.: Sensitivity of tropospheric loads and
lifetimes of short lived pollutants to fire emissions, Atmos. Chem. Phys., 15, 3543-3563,
https://doi.org/10.5194/acp-15-3543-2015, 2015.
Daskalakis, N., Tsigaridis, K., Myriokefalitakis, S., Fanourgakis, G. S., and Kanakidou, M.: Large
gain in air quality compared to an alternative anthropogenic emissions scenario, Atmos.
Chem. Phys., 16, 9771-9784, doi:10.5194/acp-16-9771-2016, 2016.
Dee, D. P., Uppala, S. M., Simmons, A. J., Berrisford, P., Poli, P., Kobayashi, S., and Vitart, F.:
The ERA-Interim reanalysis: configuration and performance of the data assimilation system,
Quarterly Journal of the Royal Meteorological Society, 137, 553–597.
http://doi.org/10.1002/qj.828, 2011.
Fotiadi, A., Hatzianastassiou, N., Drakakis, E., Matsoukas, C., Pavlakis, K. G., Hatzidimitriou, D.,
Gerasopoulos, E., Mihalopoulos, N., and Vardavas, I.: Aerosol physical and optical properties
in the eastern Mediterranean Basin, Crete, from Aerosol Robotic Network data, Atmos. Chem.
Phys., 6, 5399-5413, https://doi.org/10.5194/acp-6-5399-2006, 2006.
Gordon, H., Kirkby, J., Baltensperger, U., Bianchi, F., Breitenlecher, M., Curtius, J., Dias, A.,
Dommen, J., Donahue, N. M., Dunne, E. M., Duplissy, J., Ehrhart, S., Flagan, R. C., Frege, C.,
Fuchs, C., Hansel, A., Hoyle, C. R., Kulmala, M., Kürten, A., Lehtipalo, K., Makhmutov, V.,
Molteni, U., Rissanen, M. P., Stozhkov, Y., Tröstl, J., Tsagkogeorgas, G., Wagner, R., Williamson,
C., Wimmer, D., Winkler, P. M., Yan, C., and Carslaw, K. S.: Causes and importance of new
particle formation in the present-day and preindustrial atmospheres, J. Geophys. Res. Atmos.,
24  122, 8739-8760, 2017.

Granier, C., Guenther, A., Lamarque, J., Mieville, A., Müller, J., Olivier, J., Orlando, J., Peters,
J., Petron, G., Tyndall, G., and Wallens, S.: POET, a database of surface emissions of ozone
precursors, available at: http://www.aero.jussieu.fr/projet/ ACCENT/POET.php, 2005.
Hamed, A., Birmili, W., Joutsensaari, J., Mikkonen, S., Asmi, A., Wehner, B., Spindler, G.,
Jaatinen, A., Wiedensohler, A., Korhonen, H., Lehtinen, K. E. J., and Laaksonen, A.: Changes in
the production rate of secondary aerosol particles in Central Europe in view of decreasing $SO_2$



emissions between 1996 and 2006, Atmos. Chem. Phys., 10, 1071-1091,
https://doi.org/10.5194/acp-10-1071-2010, 2010.
Hõrrak U., Salm J., and Tammet H.: Bursts of intermediate ions in atmospheric air, J. Geophys.
Res., 103, 13909–13915, doi: 10.1029/97JD01570, 1998.
Huang, X., Zhou, L., Ding, A., Qi, X., Nie, W., Wang, M., Chi, X., Petäjä, T., Kerminen, V.-M.,
Roldin, P., Rusanen, A., Kulmala, M., and Boy, M.: Comprehensive modelling study on
observed new particle formation at the SORPES station in Nanjing, China, Atmos. Chem. Phys.,
16, 2477-2492, https://doi.org/10.5194/acp-16-2477-2016, 2016.
Inness, A., Baier, F., Benedetti, A., Bouarar, I., Chabrillat, S., Clark, H., Clerbaux, C., Coheur, P.,
Engelen, R. J., Errera, Q., Flemming, J., George, M., Granier, C., Hadji-Lazaro, J., Huijnen, V.,
Hurtmans, D., Jones, L., Kaiser, J. W., Kapsomenakis, J., Lefever, K., Leitão, J., Razinger, M.,
Richter, A., Schultz, M. G., Simmons, A. J., Suttie, M., Stein, O., Thépaut, J.-N., Thouret, V.,
Vrekoussis, M., Zerefos, C., and the MACC team: The MACC reanalysis: an 8 yr data set of
atmospheric composition, Atmos. Chem. Phys., 13, 4073-4109, https://doi.org/10.5194/acp-
15    13-4073-2013, 2013.

Kalivitis, N., Birmili, W., Stock, M., Wehner, B., Massling, A., Wiedensohler, A., Gerasopoulos,
E., and Mihalopoulos, N.: Particle size distributions in the eastern Mediterranean troposphere,
Atmos. Chem. Phys., 8, 6729-6738, doi:10.5194/acp-8-6729-2008, 2008.
Kalivitis N., Stavroulas I., Bougiatioti A., Kouvarakis G., Gagné S., Manninen H.E., Kulmala M.,
and Mihalopoulos N.: Night-time enhanced atmospheric ion concentrations in the marine
boundary layer, Atmos. Chem. Phys., 12, 3627-3638, doi:10.5194/acp-12-3627-2012, 2012.
Kalivitis, N., Kerminen, V.-M., Kouvarakis, G., Stavroulas, I., Bougiatioti, A., Nenes, A.,
Manninen, H. E., Petäjä, T., Kulmala, M., and Mihalopoulos, N.: Atmospheric new particle
formation as a source of CCN in the eastern Mediterranean marine boundary layer, Atmos.
Chem. Phys., 15, 9203-9215, doi:10.5194/acp-15-9203-2015, 2015.
Kalkavouras, P., Bossioli, E., Bezantakos, S., Bougiatioti, A., Kalivitis, N., Stavroulas, I.,
Kouvarakis, G., Protonotariou, A. P., Dandou, A., Biskos, G., Mihalopoulos, N., Nenes, A., and
Tombrou, M.: New particle formation in the southern Aegean Sea during the Etesians:
importance for CCN production and cloud droplet number, Atmos. Chem. Phys., 17, 175-192,
https://doi.org/10.5194/acp-17-175-2017, 2017.



Kerminen, V.-M. and Kulmala, M.: Analytical formulae connecting the "real" and the
"apparent" nucleation rate and the nuclei number concentration for atmospheric nucleation
events, J. Aerosol Sci., 33, 609-622, 2002.
Kerminen, V.-M., Paramonov, M., Anttila, T., Riipinen, I., Fountoukis, C., Korhonen, H., Asmi,
E., Laakso, L., Lihavainen, H., Swietlicki, E., Svenningsson, B., Asmi, A., Pandis, S. N., Kulmala,
M., and Petäjä, T.: Cloud condensation nuclei production associated with atmospheric
nucleation: a synthesis based on existing literature and new results, Atmos. Chem. Phys., 12,
12037– 12059, doi:10.5194/acp-12-12037-2012, 2012.
Kopanakis, I., Chatoutsidou, S. E., Torseth, K., Glytsos, T., and Lazaridis, M.: Particle number
size distribution in the eastern Mediterranean: Formation and growth rates of ultrafine
airborne    atmospheric    particles,    Atmospheric    Environment,    77,    790–802.
http://doi.org/10.1016/j.atmosenv.2013.05.066, 2013.
Korhonen, H., Lehtinen, K. E. J., and Kulmala, M.: Multicomponent aerosol dynamics model
UHMA: model development and validation, Atmos. Chem. Phys., 4, 757-771,
https://doi.org/10.5194/acp-4-757-2004, 2004.
Kulmala, M., Vehkamäki, H., Peťaja, T., Dal Maso, M., Lauri, A., Kerminen, V.-M., Birmili, W.,
and McMurry, P. H.: Formation and growth rates of ultrafine atmospheric particles: A review
of observations, J. Aerosol Sci., 35, 143–176, 2004a.
Kulmala, M, Kerminen, V.-M., Anttila, T., Laaksonen, A. and O'Dowd, C. D: Organic aerosol
formation    via    sulphate    cluster    activation,    J.    Geophys.    Res.,    109,    4205,
doi:10.1029/2003JD003961, 2004b.
Kulmala, M., Petäjä, T., Nieminen, T., Sipilä, M., Manninen, H. E., Lehtipalo, K., Kerminen, V.-
M.: Measurement of the nucleation of atmospheric aerosol particles, Nature Protocols, 7,
1651, http://dx.doi.org/10.1038/nprot.2012.091, 2012.
Kulmala, M., Petäjä, T., Ehn, M., Thornton, J., Sipilä, M., Worsnop, D. R., and Kerminen, V.-M.:
Chemistry of atmospheric nucleation: On the recent advances on precursor characterization
and atmospheric cluster composition in connection with atmospheric new particle formation,
Annu. Rev. Phys. Chem., 65, 21-37, 2014.



Kulmala, M., Kerminen, V.-M., Petäjä, T., Ding, A. J., and Wang L.: Atmospheric gas-to-particle
conversion: why NPF events are observed in megacities?, Faraday Discuss., 200, 271-288,
doi:10.1039/c6fd00257a, 2017.
Kyrö, E.-M., Väänänen, R., Kerminen, V.-M., Virkkula, A., Petäjä, T., Asmi, A., Dal Maso, M.,
Nieminen, T., Juhola, S., Shcherbinin, A., Riipinen, I., Lehtipalo, K., Keronen, P., Aalto, P. P.,
Hari, P., and Kulmala, M.: Trends in new particle formation in eastern Lapland, Finland: effect
of decreasing sulfur emissions from Kola Peninsula, Atmos. Chem. Phys., 14, 4383-4396,
https://doi.org/10.5194/acp-14-4383-2014, 2014.
Lazaridis, M., K. Eleftheriadis, J. Smolik, I. Colbeck, G. Kallos, Y. Drossinos, V. Zdimal, Z. Vecera,
N. Mihalopoulos, P. Mikuska, C. Bryant, C. Housiadas, A. Spyridaki, M. Astitha and V. Havranek:
Dynamics of fine particles and photo-oxidants in the eastern Mediterranean (SUB-AERO),
Atmospheric       Environment,       40,       P       6214-6228,
http://dx.doi.org/10.1016/j.atmosenv.2005.06.050, 2006.
Lehtipalo, K., Rondo, L., Kontkanen, J., Schobesberger, S., Jokinen, T., Sarnela, N., Kürten, A.,
Ehrhart, S., Franchin, A., Nieminen, T., Riccobono, F., Sipilä, M., Yli-Juuti, T., Duplissy, J.,
Adamov, A., Ahlm, L., Almeida, J., Amorim, A., Bianchi, F., Breitenlechner, M., Dommen, J.,
Downard, A. J., Dunne, E. M., Flagan, R. C., Guida, R., Hakala, J., Hansel, A., Jud, W.,
Kangasluoma, J., Kerminen, V.-M., Keskinen, H., Kim, J., Kirkby, J., Kupc, A., Kupiainen-Määttä,
O., Laaksonen, A., Lawler, M. J., Leiminger, M., Mathot, S., Olenius, T., Ortega, I. K., Onnela,
A., Petäjä, T., Praplan, A., Rissanen, M. P., Ruuskanen, T., Santos, F. D., Schallhart, S.,
Schnitzhofer, R., Simon, M., Smith, J. N., Tröstl, J., Tsagkogeorgas, G., Tomé, A., Vaattovaara,
P., Vehkamäki, H., Vrtala, A. E., Wagner, P. E., Williamson, C., Wimmer, D., Winkler, P. M.,
Virtanen, A., Donahue, N. M., Carslaw, K. S., Baltensperger, U., Riipinen, I., Curtius, J.,
Worsnop, D. R., and Kulmala, M.: The effect of acid-base clustering and ions on the growth of
atmospheric nano-particles, Nat. Commun., 7, 11594, doi:10.1038/ncomms11594, 2016.
Leino, K., Nieminen, T., Manninen, H.E., Petäjä, T., Kerminen, V.-M., and Kulmala, M.:
Intermediate ions as a strong indicator for new particle formation bursts in a boreal forest.
Boreal Env. Res. 21: 274–286, 2016.
Lelieveld, J., Berresheim, H., Borrmann, S., Crutzen, P., Dentener, F., Fischer, H., Feichter, J.,
Flatau, P., Heland, J., Holzinger, R., Korrmann, R., Lawrence, M., Levin, Z., Markowicz, K.,
Mihalopoulos, N., Minikin, A., Ramanathan, V., de Reus, M., Roelofs, G., Scheeren, H., Sciare,
J., Schlager, H., Schultz, M., Siegmund, P., Steil, B., Stephanou, E., Stier, P., Traub, M., Warneke,



C., Williams, J., and Ziereis, H.: Global air pollution crossroads over the Mediterranean,
Science, 298, 794–799, doi: 10.1126/science.1075457, 2002.
Levelt, P. F., van den Oord, G. H. J., Dobber, M. R., Malkki, A., Visser, H., de Vries, J., Stammes,
P., Lundell, J. O. V., and Saari, H.: The Ozone Monitoring Instrument, IEEE T. Geosci. Remote,
44, 1093–1101, doi: 10.1109/TGRS.2006.872333, 2006.
Madronich, S.: The atmosphere and UV-B radiation at ground level. Environmental UV
Photobiology, Plenum Press, 1–39, 1993.
Manninen, H.E., Nieminen, T., Asmi, E., Gagné, S., Häkkinen, S., Lehtipalo, K., Aalto, P., Kivekäs
,N., Vana, M., Mirme, A., Mirme, S., Hõrrak, U., Plass-Dülmer, C., Stange, G., Kiss, G., Hoffer,
A., Moerman, M., Henzing, B., Brinkenberg, M., Kouvarakis, G.N., Bougiatioti, K., O'Dowd, C.,
Ceburnis, D., Arneth, A., Svenningsson, B., Swietlicki, E., Tarozzi, L., Decesari, S., Sonntag, A.,
Birmili, W., Wiedensohler, A., Boulon, J., Sellegri, K., Laj, P., Baltensperger, U., Laaksonen, A.,
Joutsensaari, J., Petäjä, T., Kerminen, V.-M., and Kulmala M.: EUCAARI ion spectrometer
measurements at 12 European sites – analysis of new particle formation events, Atmos. Chem.
Phys., 10, 7907-7927, 2010.
Mihalopoulos, N., Stephanou, E., Kanakidou, M., Pilitsidis, S., and Bousquet, P.: Tropospheric
aerosol ionic composition in the eastern Mediterranean region, Tellus Series B - Chemical and
Physical Meteorology, 49, 314– 326, 1997.
Mogensen, D., Gierens, R., Crowley, J. N., Keronen, P., Smolander, S., Sogachev, A., Nölscher,
A. C., Zhou, L., Kulmala, M., Tang, M. J., Williams, J., and Boy, M.: Simulations of atmospheric
OH, $O_3$ and $NO_3$ reactivities within and above the boreal forest, Atmos. Chem. Phys., 15, 3909-
3932, https://doi.org/10.5194/acp-15-3909-2015, 2015.
Myriokefalitakis,S., Vignati, E., Tsigaridis, K., Papadimas, C., Sciare, J.,  Mihalopoulos, N.,
Facchini, M. C., Rinaldi, M., Dentener, F. J., Ceburnis, D.,  Hatzianastasiou, N., O'Dowd, C.D.,
van Weele, M., and Kanakidou, M.: Global modelling of the oceanic source of organic aerosols,
Advances in Meteorology, doi:10.1155/2010/939171, 2010.
Myriokefalitakis S., Daskalakis, N., Fanourgakis, G. S., Voulgarakis, A., Krol, M. C., Aan de Brugh,
J. M. J.,  and Kanakidou, M.: Pollution over the Mediterranean Basin: The Importance of Long-
Range Transport on ozone and carbon monoxide, Science of The Total Environment,563–564,

30  40, 2016.



Nieminen, T., Asmi, A., Dal Maso, M., Aalto, P. P., Keronen, P., Petäjä, T., Kulmala, M., and
Kerminen, V.-M.: Trends in atmospheric new-particle formation: 16 years of observations in a
boreal-forest environment, Boreal Env. Res., 19, 191-214, 2014
O'Dowd, C. D., Hämeri, K., Mäkelä, J. M., Pirjola, L., Kulmala, M., Jennings, S. G., Berresheim,
H., Hansson, H.-C., de Leeuw, G., Kunz, G. J., Allen, A. G., Hewitt, C. N., Jackson, A., Viisanen,
Y., and Hoffmann, T.: A dedicated study of New Particle Formation and Fate in the Coastal
Environment (PARFORCE): Overview of objectives and achievements, J. Geophys. Res., 107,
8108, doi:10.1029/2001jd000555, 2002.
Paraskevopoulou, D., Liakakou, E., Gerasopoulos, E., and Mihalopoulos, N.: Sources of
atmospheric aerosol from long-term measurements (5 years) of chemical composition in
Athens, Greece. Science of The Total Environment, 527–528, 165–178.
http://doi.org/10.1016/J.SCITOTENV.2015.04.022, 2015.
Petäjä, T., Kerminen, V.-M., Dal Maso, M., Junninen, H., Koponen, I.K., Hussein, T., Aalto, P.P.,
Andronopoulos, S., Robin, D., Hämeri, K., Bartzis, J.G. and Kulmala, M.: Sub-micron
atmospheric aerosols in the surroundings of Marseille and Athens: physical characterization
and new particle formation, Atmos. Chem. Phys., 7, pp. 2705-2720, doi:10.5194/acp-7-2705-

17  2007, 2007.

Pikridas, M., Riipinen, I., Hildebrandt, L., Kostenidou, E., Manninen, H., Mihalopoulos, N.,
Kalivitis, N., Burkhart, J., Stohl, A., Kulmala, M. and Pandis, S. N.: NPF at a remote site in the
eastern Mediterranean, J. Geophys. Res., 117, D12205, doi:10.1029/2012JD017570, 2012.
Sciare, J., Bardouki, H., Moulin, C., and Mihalopoulos, N.: Aerosol sources and their
contribution to the chemical composition of aerosols in the eastern Mediterranean Sea during
summertime, Atmos. Chem. Phys., 3, 291-302, https://doi.org/10.5194/acp-3-291-2003,

24  2003.

Salimi, F., Rahman, M. M., Clifford, S., Ristovski, Z., and Morawska, L.: Nocturnal new particle
formation events in urban environments. Atmos. Chem. Phys., 17, 521–530.
http://doi.org/10.5194/acp-17-521-2017, 2017.
Siakavaras, D., Samara, C., Petrakakis, M., and Biskos, G.: Nucleation events at a coastal city
during the warm period: Kerbside versus urban background measurements. Atmospheric
Environment, 140, 60–68. http://doi.org/10.1016/j.atmosenv.2016.05.054, 2016.



Spracklen, D. V., Carslaw, K. S., Kulmala, M., Kerminen, V.-M., Mann, G. W., and Sihto, S.-L.:
The contribution of boundary layer nucleation events to total particle concentrations on
regional and global scales, Atmos. Chem. Phys., 6, 5631-5648, doi:10.5194/acp-6-5631-2006,

4 2006.

Tammet, H., Komsaare, K., and Hõrrak, U.: Intermediate ions in the atmosphere, Atmospheric
Research,135–136, 263–273,http://doi.org/https://doi.org/10.1016/j.atmosres.2012.09.009,

7 2014.

Tröstl, J., Chuang, W. K., Gordon, H., Heinritzi, M., Yan, C., Molteni, U., Ahlm, L., Frege, C.,
Bianchi, F., Wagner, R. and Simon, M., Lehtipalo, K., Williamson, C., Craven, J. S., Duplissy, J.,
Adamov, A., Almeida, J., Bernhammer, A.-K., Breitenlechner, M., Brilke, S., Dias, A., Ehrhart,
S., Flagan, R. C., Franchin, A., Fuchs, C., Guida, R., Gysel, M., Hansel, A., Hoyle, C. R., Jokinen,
T., Junninen, H., Kangasluoma, J., Keskinen, H., Kim, J., Krapf, M., Kürten, A., Laaksonen, A.,
Lawler, M., Leiminger, M., Mathot, S., Möhler, O., Nieminen, T., Onnela, A., Petäjä, T., Piel, F.
M., Miettinen, P., Rissanen, M. P., Rondo, L., Sarnela, N., Schobesberger, S., Sengupta, K.,
Sipilä, M., Smith, J. N., Steiner, G., Tomè, A., Virtanen, A., Wagner, A. C., Weingartner, E.,
Wimmer, D., Winkler, P. M., Ye, P., Carslaw, K. S., Curtius, J., Dommen, J., Kirkby, J., Kulmala,
M., Riipinen, I., Worsnop, D. R., Donahue, N. M., and Baltensperger, U.: The role of low-
volatility organic compounds in initial particle growth in the atmosphere, Nature, 533, 527–

19 531, 2016.

Tzitzikalaki, E., Kalivitis, N., Kouvarakis, G., Daskalakis, N., Kerminen, V.-M., Mihalopoulos, N.,
Boy, M., and Kanakidou, M.: Simulations of New Particle Formation and Growth Processes at
eastern Mediterranean, with the MALTE-Box Model, in: Perspectives on Atmospheric
Sciences. T. Karacostas, A. Bais, & P. T. Nastos (Eds.), (pp. 933–939). Cham: Springer
International Publishing, 2017.
Wang, Z., Wu, Z., Yue, D., Shang, D., Guo, S., Sun, J., Ding, A., Wang, L., Jiang, J., Guo, H., Gao,
J., Cheung, H. C., Morawska, L., Keywood, M., and Hu, M.: New particle formation in China:
Current knowledge and further directions, Sci. Total Environ., 577, 258-266,
http://dx.doi.org/10.1016/j.scitotenv.2016.10.177, 2017.
Wiedensohler, A., Birmili, W., Nowak, A., Sonntag, A., Weinhold, K., Merkel, M., Wehner, B.,
Tuch, T., Pfeifer, S., Fiebig, M., Fjäraa, A. M., Asmi, E., Sellegri, K., Depuy, R., Venzac, H., Villani,
P., Laj, P., Aalto, P., Ogren, J. A., Swietlicki, E., Williams, P., Roldin, P., Quincey, P., Hüglin, C.,
Fierz-Schmidhauser, R., Gysel, M., Weingartner, E., Riccobono, F., Santos, S., Grüning, C.,



Faloon, K., Beddows, D., Harrison, R., Monahan, C., Jennings, S. G., O'Dowd, C. D., Marinoni,
A., Horn, H.-G., Keck, L., Jiang, J., Scheckman, J., McMurry, P. H., Deng, Z., Zhao, C. S.,
Moerman, M., Henzing, B., de Leeuw, G., Löschau, G., and Bastian, S.: Mobility particle size
spectrometers: harmonization of technical standards and data structure to facilitate high
quality long-term observations of atmospheric particle number size distributions, Atmos.
Meas. Tech., 5, 657-685, doi:10.5194/amt-5-657-2012, 2012.





7) Tables

| Day classification | Number of events | % |
|---|---|---|
| Total events | 623 | 29.37 |
| Class I | 161 | 7.59 |
| Class II | 462 | 21.78 |
| | | |
| Undefined | 555 | 26.17 |
| Non-event | 943 | 44.46 |
| | | |
| Total days | 2121 | 100.00 |

Table 1) Total available measurement days and percentage of NPF events observed at
Finokalia during the period June 2008-June 2015

| | $J_9$ ($cm^{-3} s^{-1}$) | | | $GR_{9-25}$ ($nm\ hr^{-1}$) | | | $CS \times 10^{-3}$ ($s^{-1}$) | | |
|---|---|---|---|---|---|---|---|---|---|
| | Mean | Median | SD | Mean | Median | SD | Mean | Median | SD |
| Winter | 0.7 | 0.6 | 0.4 | 3.3 | 2.7 | 2.2 | 3.7 | 3.1 | 2.6 |
| Spring | 0.9 | 0.8 | 0.6 | 3.5 | 2.7 | 2.5 | 5.5 | 5.1 | 2.7 |
| Summer | 0.7 | 0.6 | 0.4 | 6.8 | 6.4 | 3.5 | 9.1 | 8.9 | 3.7 |
| Autumn | 1.0 | 0.9 | 0.9 | 5.0 | 4.4 | 2.9 | 6.3 | 5.8 | 3.6 |

Table 2) Formation rates for 9nm particles ($J_9$), growth rates in the size range 9-25 nm ($GR_{9-25}$)
for NPF events observed at Finokalia and condensational sink for sulphuric acid (CS) on
seasonal base during the period June 2008-June 2015 (mean, median and standard deviation).


8) Figures

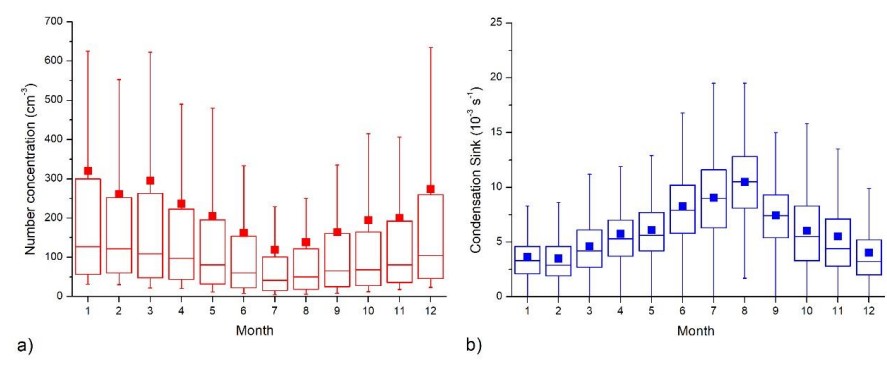

a)                                                     b)
1) Monthly average variation of a) nucleation mode particle number concentration and b)
sulfuric acid condensational sink (CS) at Finokalia station over the period June 2008-June 2015.
Whiskers represent 10th and 90th percentiles, box edges are 75th and 25th percentiles, the
line in the box is the median, the solid square is the mean.

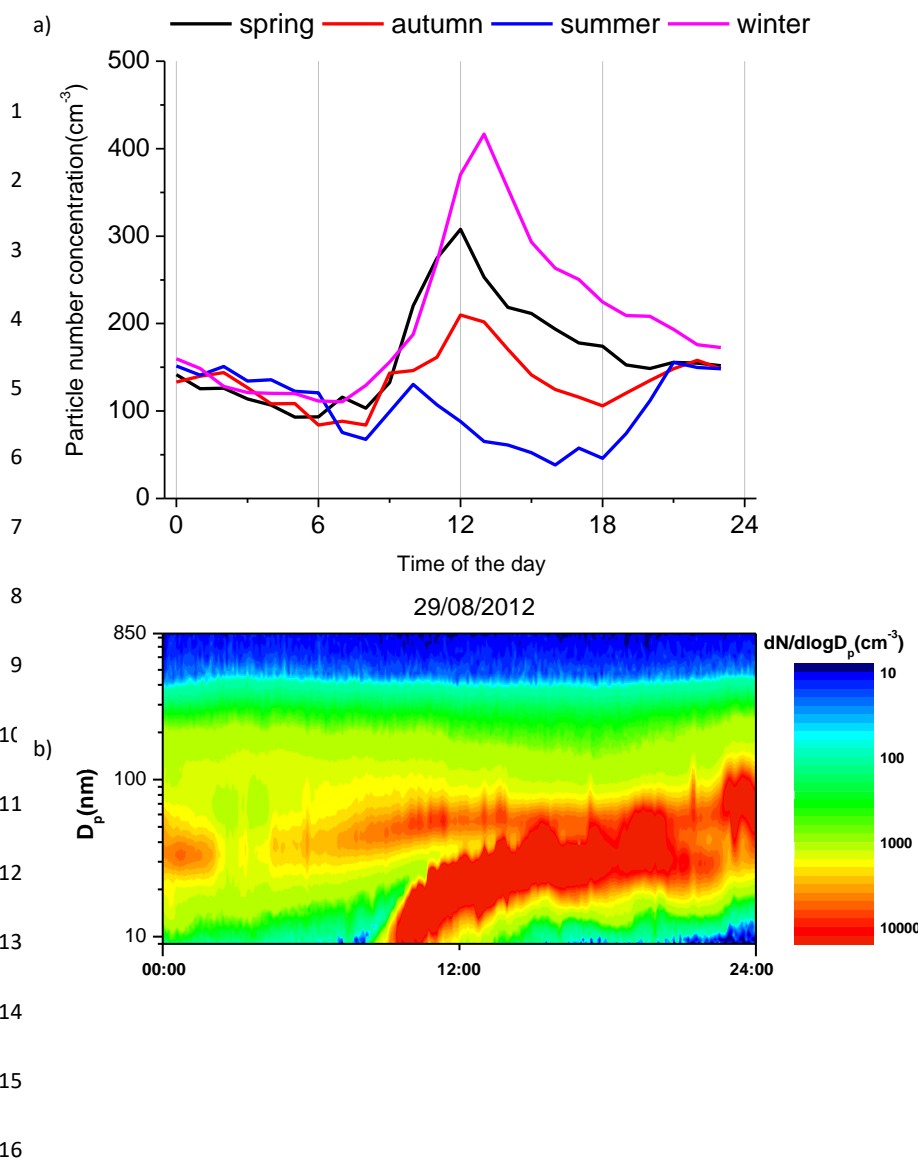

2) a) Average diurnal variation of nucleation mode particle number concentration (hourly values) at Finokalia over the period June 2008-June 2015 b) New particle formation event captured at Finokalia on 29/08/2012



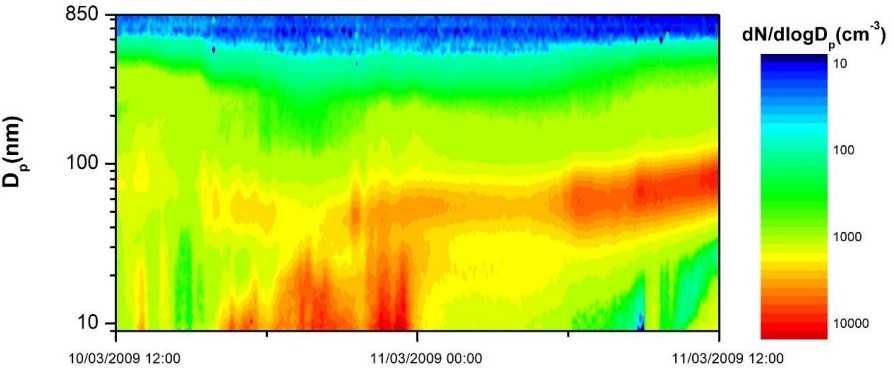

2    3) Example of appearance of nucleation mode particles during several hours as observed

3    during the night of 10 to 11/09/2009.



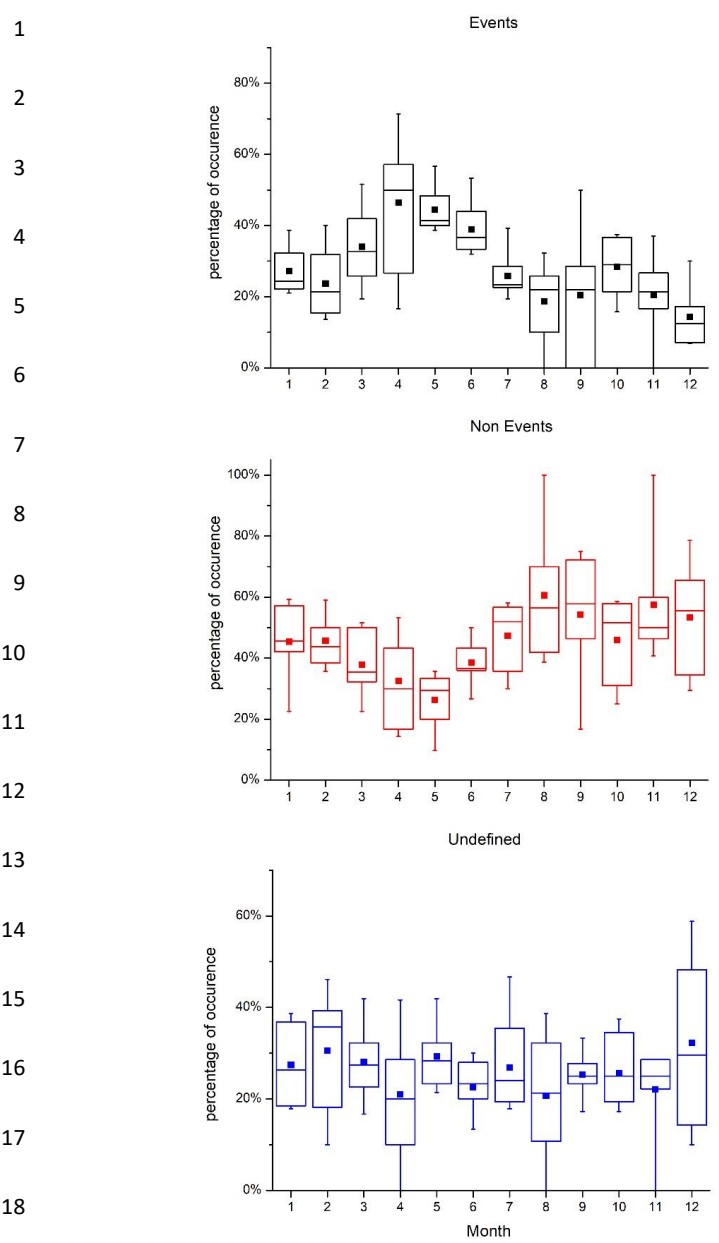

4) Seasonal variation of NPF percentage of occurrence of event, non-event and undefined days

relatively to available measurement days at Finokalia for the period June 2008-June2015.

Whiskers represent 10th and 90th percentiles, box edges are 75th and 25th percentiles, the

line in the box is the median, square is mean.




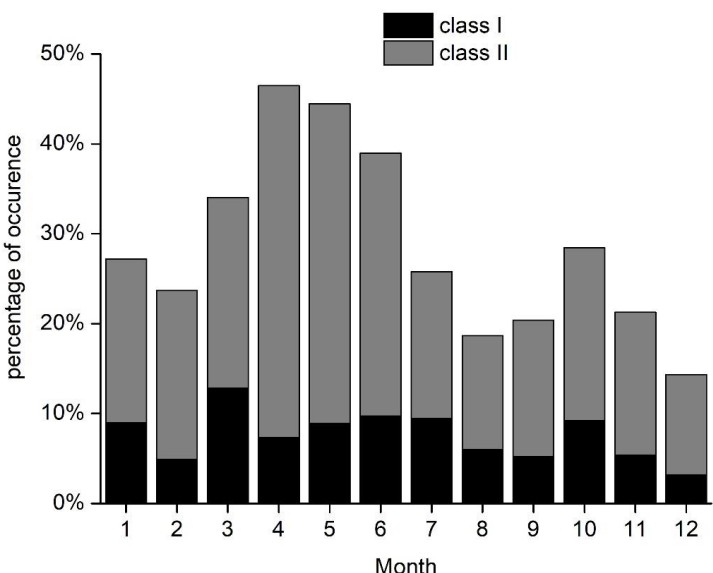

2  5) Seasonal variation of percentage of occurrence of NPF Class I & II events relatively to

3  available measurement days at Finokalia in the eastern Mediterranean for the period June

4  2008-June2015.



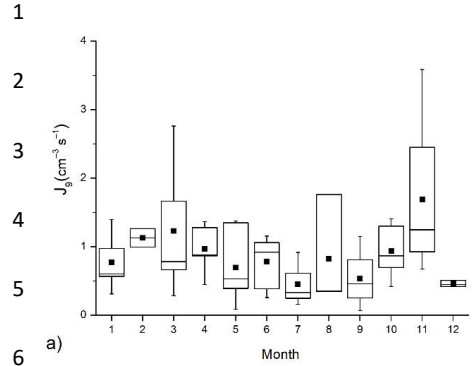 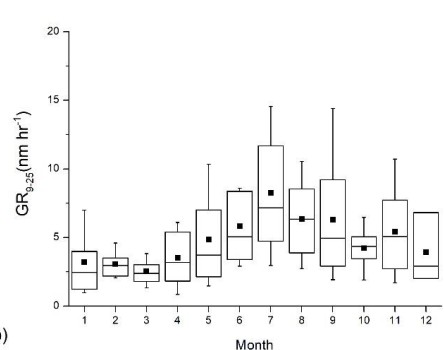

a)                                    b)

6) Seasonal variation of a) formation rate of 9nm particles ($J_9$) and b) growth rate in the size

range 9-25nm ($GR_{9-25}$) as calculated during Class I NPF events at Finokalia for the period June

2008-June 2015. Whiskers represent 10th and 90th percentiles, box edges are 75th and 25th

percentiles, the line in the box is the median and the solid square is the mean.




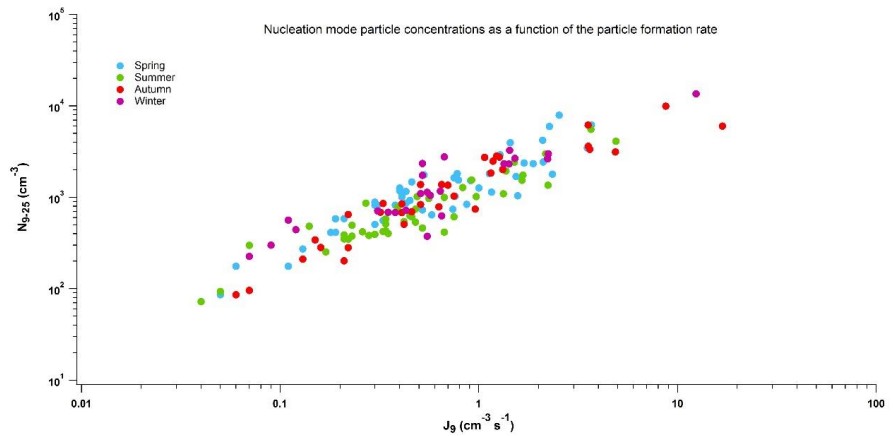

2    7) Scatter plot of formation rate of 9nm particles ($J_9$) versus the number concentration of

3    nucleation mode particles ($N_{9-25}$) (hourly maximum value during the event) at Finokalia, for

4    events that $J_9$ could be calculated with a good level of confidence (Class I events).



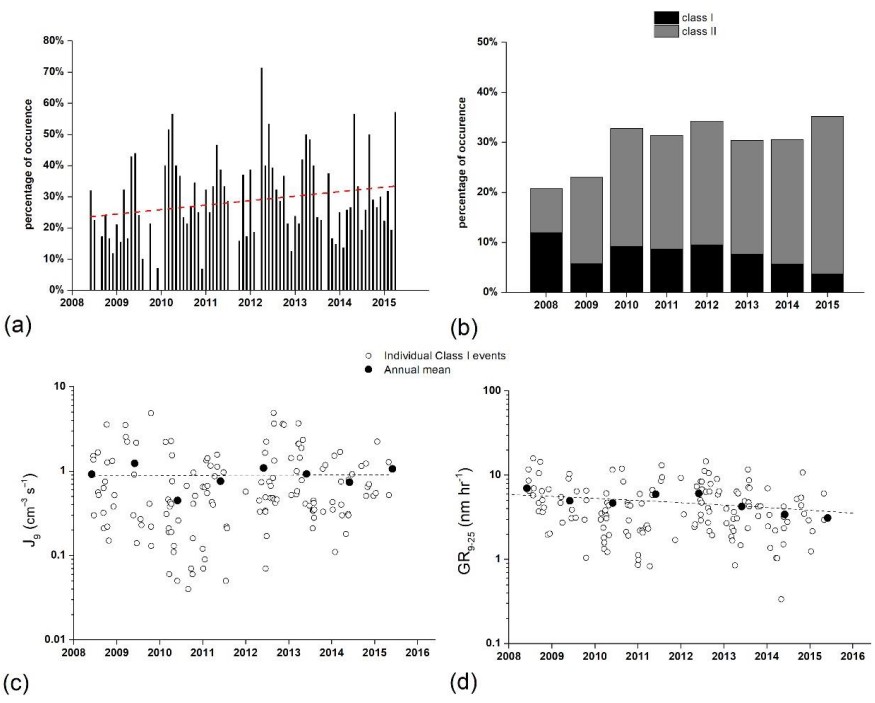

8) a) Time series of monthly NPF percentage of occurrence at Finokalia for the years 2008-
2015. b) Annual NPF percentage of occurrence at Finokalia for the period June 2008-June 2015
for Class I&II events. Interannual variation of c) formation rates of 9nm particles ($J_9$) and d)
growth rate in the size range 9-25nm ($GR_{9-25}$) during Class I NPF events at Finokalia for the
period June 2008-June 2015 (solid circles represent annual averages).

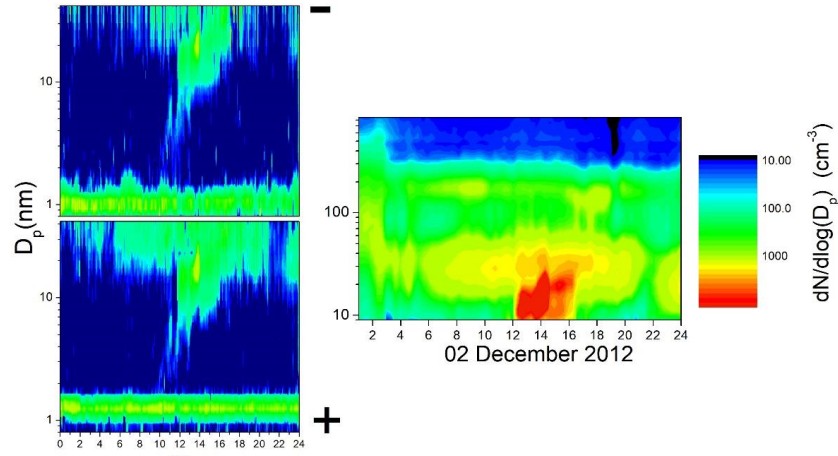

6    9) Nucleation event observed at Finokalia on 2 December 2012 as captured by AIS (left panels

7    for negative (up) and positive (bottom) polarity) and SMPS (right panel)



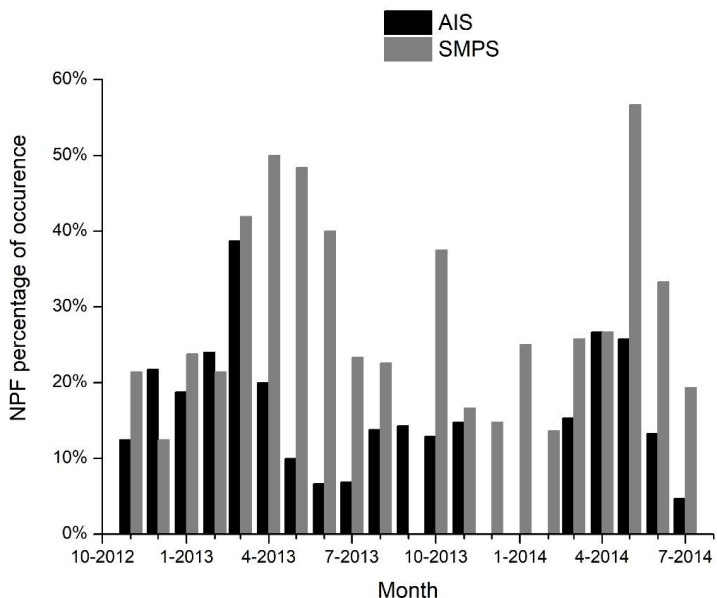

10) Monthly variability of NPF events' percentage of occurrence relatively to available
measurement days at Finokalia as determined by analysis of AIS data during the FRONT
experiment (Nov. 2012-July 2014). For a direct comparison, the monthly variability of NPF
events as obtained from the SMPS measurements for the same period is included.



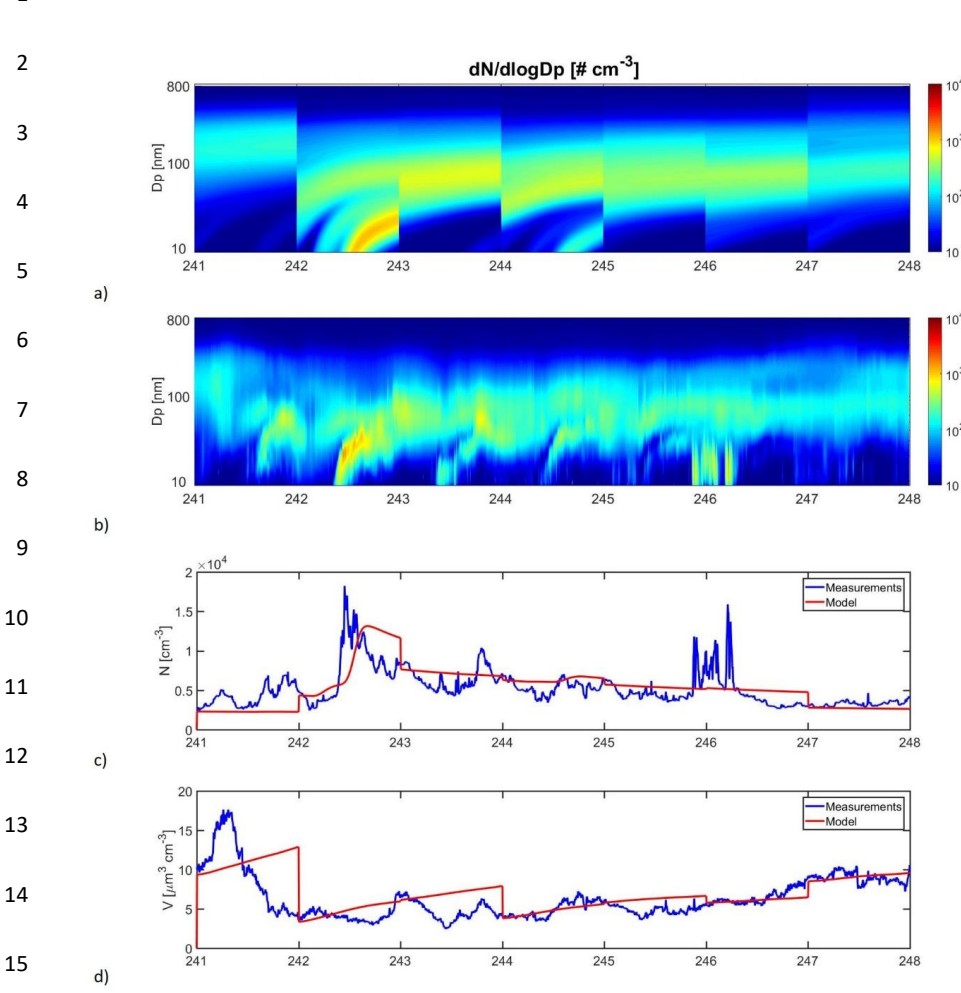

a)

b)

c)

d)

11) Particle size distributions at Finokalia for the event week: a) simulations with the adjusted parameters for the sub-tropical environment The discontinuities observed every midnight in the model results are due to initialization of the model every midnight with measured number size distributions. b) observations of number size distributions (modified from Tzitzikalaki et al., 2017). Measured and modelled d) total number concentration and e) total volume concentration for the same period. The x-axis in all figures is Julian day of 2012.





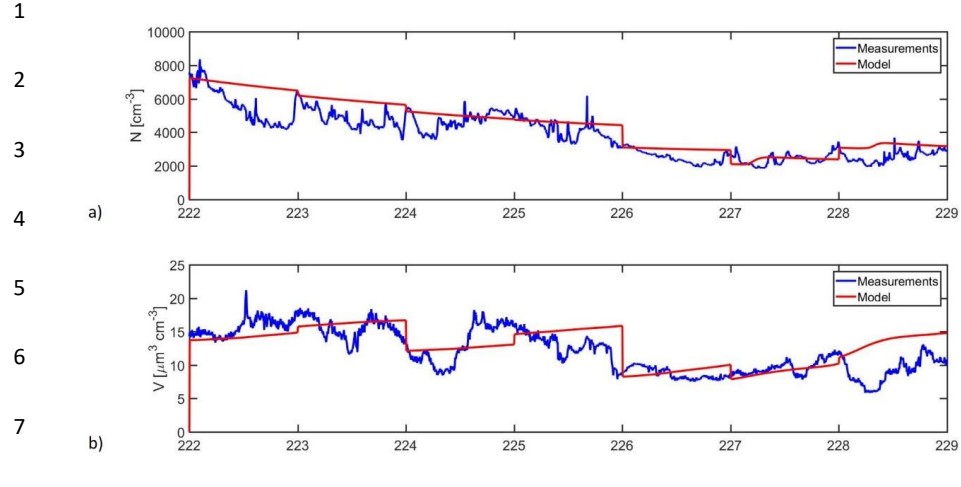

12) Measured and modelled a) total number concentration and b) total volume
concentration for a non-event week at Finokalia. The x-axis in both figures is Julian day of
11 2012.

