# Peer review of "Formation and growth of atmospheric nanoparticles in the eastern Mediterranean: Results"

_Atmospheric Chemistry and Physics, 2018_

## Referee Comment (RC1) · Anonymous Referee #1 · 16 May 2018

The paper by Kalivitis et al. presents long term measurement of particle size distribution from Finokalia (eastern Mediterranean region). The main focus of the study is on nucleation mode particles and characteristics of new particle formation (NPF) events, including frequency of occurrence as well as particle formation and growth rates. The last part of the paper is dedicated to a simulation case study of NPF with the MALTE-box model.

I recommend the publication of this paper, as it is well written and describes a valuable dataset which allows for the investigation of NPF over 7 years, thus contributing to our understanding of the process. I would however suggest some revisions before final publication of this study. In particular, some of the observations/conclusions reported throughout the manuscript should be slightly balanced. Also, I am not fully convinced by the modelling part in its current form: it is in my view missing a clear presentation of the strategy/sensitivity tests which lead to the final "good simulation", and it would also benefit from a quick discussion on the relevance of the values finally used for some of the key variables (e.g. monoterpenes concentration). Moreover, it is not clear to me how the analysis reported in Section 3.5 of the present paper differs from that of Tzitzikalaki et al. (2017), as I cannot access this source. Detailed comments are listed below.

P3, L13-16 : I would suggest to clearly mention "only when accumulation mode particles were neutral", as with the current form of the sentence it is a bit confusing whether those particles are pre-existing particles or the newly formed ones.

P4, L21: I would suggest to remove "from the early stages of nucleation", since I think those cannot be investigated when measuring particles larger than 9 nm. Such statement would better suit to AIS measurements or to measurements conducted with instruments such as the particle size magnifier (PSM, Vanhanen et al., 2011), which allows for the detection of ~1-1.5 nm particles (charged + neutral).

P5, L9-10: Please refer the reader to Mirme et al., (2007) for AIS measurements. Also, could the authors give more information about the uncertainties reported on L14-15 (calculation method or reference to a paper)?

P5, L22-31: Several short/minor comments about the description of the calculations:

- L22: instead of "particles with diameter D" (should at least be $D_p$) and since the formation rate is not calculated for different particle diameters, I would clearly mention $D_p$ = 9nm, otherwise one has to wait until Section 3.2 to explicitly get this information (and it would also be more consistent with the description of the terms of Eq. (1));
- L25: "CoagS is the coagulation of particles in this size range" (should be $CoagS_{D_p}$): which (lowest) particle size was used to calculate $CoagS_{D_p}$? I would suggest a more accurate formulation, such as "$CoagS_{D_p}$ is the coagulation sink of XX nm particles on larger particles";
- L27: Please refer the reader to Dal Maso et al. (2005) for the mode fitting method;
- L31: For this first occurrence, instead of "the sulfuric acid sink", I would suggest to rather write something more explicit like "CS is the condensation sink caused by the pre-existing aerosol population and was calculated using the characteristics/properties of sulfuric acid".

P6, L11: "relevant chemical reactions": I would recommend to add few words on the relevance of the reactions, at least mention they are related to sulfuric acid production.

P6, L26-27: what is "free form nucleation"? I would suggest to briefly recall the parameterization which is used in the model and introduce the "nucleation coefficient", later discussed in Section 3.5 (P16, L16 & L24-25).

P6, L29-30: "All these compounds were treated as sulfuric acid and organics": what does this sentence mean? Also, on L27, if ELVOCs are considered please add "20 extremely low-volatility organic compounds", otherwise change to "LVOCs."

P7, L6-7: I am a bit confused with this sentence: only the particle size distributions are used to initialize the model (as reported on P6, L24-25), which then calculates a CS based on the simulated distributions, right? If the purpose of the abovementioned sentence is only to precise that SMPS data were used to calculate the CS, I would strongly recommend to move it to Section 2.1 (P5, L31), as Section 2.2 is dedicated to model description.

P8, L11-13: I am a bit confused with the use of TUV: was the parameterisation used instead of TUV, or implemented in TUV?

P8, L21-27: I am somewhat sceptical about the values which are reported in this paragraph; I think they do not give much information since the shape of the particle size distribution is highly variable with respect to seasons, event vs non-event days, time of the day… I would thus suggest to either provide a more detailed description/comparison of the concentrations in the different modes and their contribution to total concentration, or at least provide quartiles/standard deviation for all reported values (not only for nucleation mode particle concentration).

P8, L31: are the times local or UTC?

P8, L32 - P9, L1: "Such an observation suggests that the nucleation particle number concentration is controlled by NPF episodes". Isn't it what we expect by definition? Which other sources would the authors expect for particles in this size range? This comment also refers to P2 L16-17, P9 L25, P11 L21-22, P15 L22-23. Moreover, concerning the statement P11 L20-22, I am not sure if the linear relation between $J_9$ and nucleation mode particle concentration ($N_{9-25}$) can be considered as a strong support for NPF being the main source of nucleation particle, since according to Eq. (1) $J_9$ calculation includes $N_{9-25}$ in two of the three terms.

P9, L10-23: I think that even if deep investigation of night time events is not in the scope of this paper, slightly more detailed description could be provided. In specific:

- L10-12: Even if similar night time concentrations are observed during all seasons, they seem to result from different processes based on Fig. 2a. Indeed, there is an increase of the concentrations after 18:00 in summer and autumn, which may suggest evening time new particle formation, but during spring and winter the concentrations keep on decreasing until they reach the night time value, suggesting that evening events are not frequent during these seasons. I would thus suggest to balance the sentence from L10-12, and maybe provide frequencies of occurrence of such events for each season, which will also help quantifying "Frequently" (L13).
- L17-18: I would also add that on top of the "local" character of these events, which may partly explain the limited source of condensing vapours (and therefore particle growth), the absence of photochemistry during night time most likely strengthen the lack of vapours needed to sustain particle growth.

P10, L3-18: I wouldn't say that ozone is "the major oxidant in the atmosphere", especially when focussing on daytime NPF events, during which OH is expected to play a significant (major?) role. Also, I don't think that based on the variables included in this factor analysis it is possible to state that NPF is not sensitive to "atmospheric chemical composition"; compounds other than ozone such as for e.g. NOx, SO₂, monoterpenes… would be needed to draw such conclusions.

P11, L3: "the particle survival probability seems to be the highest in winter": the authors have the data needed to actually test their hypothesis and provide a more robust conclusion, and even quantify the variations of the survival probability in different seasons.

P11: While they peak at slightly different times of the year, the maximum of the NPF frequency, particle formation and growth rates are all attributed to enhanced biogenic emissions and/or photochemistry (P10 L27, P11 L16-17 and P11 L25-26, respectively). This hypothesis seems plausible as all maxima are observed during spring/summer, but could the authors comment on the different seasonal variations

of the abovementioned variables? In contrast it can be seen from Fig. 1a and 6b that the GR and CS have similar seasonal patterns: is it then realistic to think that CS and the vapours involved in particle growth "share the same origin"?

P11, L27-28: It is true that based on Fig 2a the average duration of NPF in summer seems to be shorter compared to other seasons, but also the maximum of the concentration is lower. Since the CS (and consequently CoagS) is also higher during summer (Fig. 1a), I would think that both the CS (CoagS) and the GR are affecting the variation of nucleation mode particle concentration (should be checked by calculating the survival probability).

P12, L1, L5: I would slightly balance the statements ("notable increase", "clear decreasing") as in my opinion the reported observations are not as obvious as suggested.

P12, L5-33: I am not fully convinced by the conclusion reported on L30, which suggests that decreased $SO_2$ concentrations related to the economic crisis in Europe may explain observed variations of GR and occurrence of class I NPF events. Main reasons for this are listed below:

- The lack of $SO_2$ measurement in Finokalia prevents from any direct evaluation of the $SO_2$ concentration decrease at this site;
- Based on previous studies mentioned in the present work it seems that decreasing $SO_2$ concentrations can lead to contrasting observations, thus pointing to the fact that robust conclusions cannot be inferred from the analysis of $SO_2$ alone;
- While the important role of $H_2SO_4$ in early nucleation stage has been reported in different studies, the need for other species to explain observed GR has also been evidenced, and the present paper itself tends to emphasize the role of organic species in NPF at Finokalia. Indeed, maximum of NPF occurrence, $J_9$ and GR are all attributed to enhanced biogenic emissions, and best agreement between model simulation and observation is achieved when adjusting monoterpenes concentration in the model. I would thus think that $SO_2$ driving the observed variations of GR and NPF occurrence is not fully consistent with the aforementioned observations/results.

P13, L12-14: Does this sentence mean that instrument malfunction was affecting measurement of positive ions? If not, it is fine to focus on negative ions only, but I wouldn't justify this choice based on their better ability to represent NPF events. Indeed, it is in my opinion complex to assess which polarity gives the "better representation of NPF events", as the different observations from the two DMAs may instead reflect the signature of the nucleation mechanism.

P13, L17-26: I would have expected the AIS-derived NPF frequencies to be more often higher (or at least equal) than SMPS-derived ones, while the opposite is shown on Fig. 10. Does it mean that the event day illustrated on Fig. 9 is only representative of a rather limited fraction of the events observed in Finokalia, while the majority of them is actually not visible from the AIS smallest diameters? In order to make the most of the FRONT dataset and provide more information on the nature of the events detected in Finokalia during this period, I would suggest to also report for each month (on Fig. 10 for instance) the number of events detected by each instrument and the number of event days they have in common. This will help assessing the fraction of events with very limited growth only visible in AIS data, the fraction of regional events detected by both instruments and that of events only visible in SMPS data.

P14, L6-33: I have several comments/questions regarding model simulations:

- L15: What does "NPF level" mean?
- L22-26: Could the authors briefly summarize the strategy they adopted to finally reach fair agreement between model simulation and observation? For instance, which sensitivity tests

were performed, were parameters other than nucleation coefficient and monoterpenes concentration also tuned?

- L25: are the levels of final simulated monoterpenes concentration realistic, are they for instance in agreement with observations from 2014?
- L27-29: I would slightly balance the conclusions ("well captured", "in such detail"), as if I agree with the fact that the reported results are very encouraging, one can observe some discrepancies between model and observation (e.g. NPF event from day 243 in not visible in model data);
- L29-31: Do the authors also consider the possibility to test other nucleation mechanisms in future simulations?

References:

Mirme, A., E. Tamm, G. Mordas, M. Vana, J. Uin, S. Mirme, T. Bernotas, L. Laakso, A. Hirsikko, and M. Kulmala: A wide-range multi-channel Air Ion Spectrometer, Boreal Environ. Res., 12, 247–264, 2007.

Vanhanen, J., Mikkilä, J., Lehtipalo, K., Sipilä, M., Manninen, H. E., Siivola, E., Petäjä, T., and Kulmala, M.: Particle size magnifier for Nano-CN detection, Aerosol Sci. Tech., 45, 533–542, 2011.

---

## Referee Comment (RC2) · Anonymous Referee #2 · 5 Jul 2018

"Formation and growth of atmospheric nanoparticles in the eastern Mediterranean: Results from long-term measurements and process simulations" reports long term data from a Station in South Europe. Whilst the data are of good quality and worth publication (long term smps data are scarce) the analysis is fair and does not add any new result. The increasing trend (and decreasing trend) or NPF and GR (respectively) may suffer from lack of data at the beginning of the period (2008-2012) relative to the last part of the period (2012-2015) - the trend may be a simple artifact.

- line 8 abstract : biogenic - marine or land or both? specify - line 11-13 Do not un-

derstand what the sentence means. please reprhase and specify simply you see NPF during night time (seen elasewhere too). - sentence 18-22 not very clear - maybe concomitant ion size distributions suggests . . . - pg 3 perhaps report the study of Dall'Osto et al (2018, Sci. Rep.) reported by same co-authors showing south Europe is different from Central and North Europe. - pg 8 line 16-20 I think this is not correct, likely the longest SMPS size distrubutions are likely in Barcelona and regional areas of Montsein (Dr. Querol s group). - Increase NPF events and decreased GR - this is interesting, it makes sense if the CS is lower over time, there is likely more NPF events, and they likely grow less cause likely you have less condensing material. - Figure 8a. I think the whole conclusion may simply be noise. If looking at figure 8a, you see 2008-2010 you have less datapoints (perhaps in spring - summer) that causes the trend you may have. It looks if you remove the 2008-2009, the trend to me is not existing. I would be careful to say there is a trend (and so I would remove and change all abstract) - it is visually clear that years 2008-2010 have smaller data coverage that 2013-2015. - Considering the above, I see this study does not add much additional novel results, although it is worth publication cause you clearly see a long SMPS time trend showing spring nucleations (different from typical summer ones).

―――――――――――――――――

---

## Author Response (AR1)

**1 Response to Reviewer #1**

**1** Response to reviewer #1**

-The authors would like to thank the reviewer for the comments that helped to improve this
manuscript. Please find below a point-by-point reply to all of the issues raised and the
corresponding changes

The paper by Kalivitis et al. presents long term measurement of particle size distribution from 6 Finokalia (eastern Mediterranean region). The main focus of the study is on nucleation mode 7 particles and characteristics of new particle formation (NPF) events, including frequency of 8 occurrence as well as particle formation and growth rates. The last part of the paper is dedicated 9 to a simulation case study of NPF with the MALTE-box model.

| 11 | I recommend the publication of this paper, as it is well written and describes a valuable dataset |
|----|---------------------------------------------------------------------------------------------------|
| 12 | which allows for the investigation of NPF over 7 years, thus contributing to our understanding    |
| 13 | of the process.                                                                                   |

I would however suggest some revisions before final publication of this study. In particular,
 some of the observations/conclusions reported throughout the manuscript should be slightly
 balanced.

Also, I am not fully convinced by the modelling part in its current form: it is in my view missing a clear presentation of the strategy/sensitivity tests which lead to the final "good simulation", and it would also benefit from a quick discussion on the relevance of the values finally used for some of the key variables (e.g. monoterpenes concentration).

-We have now added information in the modelling part regarding the simulation tests the led
to the adequate agreement with the observations regarding the nucleation coefficient and the
changes in the monoterpene concentrations.

Moreover, it is not clear to me how the analysis reported in Section 3.5 of the present paper 31 differs from that of Tzitzikalaki et al. (2017), as I cannot access this source.

-The Tzitzikalaki et al., 2017 publication refers to COMECAP 2016 conference proceedings
 where the contour plots of the simulations were presented and briefly described. The contour

-We have tried to balance the conclusions throughout the manuscript.

plot has been completely removed and only number and volume concentrations are now 2 presented. 3 Detailed comments are listed below. 4 5 6 P3, L13-16: I would suggest to clearly mention "only when accumulation mode particles were neutral", as with the current form of the sentence it is a bit confusing whether those particles 7 are pre-existing particles or the newly formed ones. 8 9 10 -The sentence was changed according to suggestion. 11 12 P4, L21: I would suggest to remove "from the early stages of nucleation", since I think those 13 cannot be investigated when measuring particles larger than 9 nm. Such statement would better 14 suit to AIS measurements or to measurements conducted with instruments such as the particle 15 size magnifier (PSM, Vanhanen et al., 2011), which allows for the detection of ~1-1.5 nm 16 particles (charged + neutral). 17 18 -The sentence was changed according to suggestion as the SMPS operated at Finokalia can 19 measure particles larger than 9 nm. 20 21 P5, L9-10: Please refer the reader to Mirme et al., (2007) for AIS measurements. Also, could 22 the authors give more information about the uncertainties reported on L14-15 (calculation 23 method or reference to a paper)? 24 25 -The reference Mirme et al., 2007 was added and for more information for the calibration and 26 uncertainties of AIS we added the reference to Manninen et al, 2010. 27 28 P5, L22-31: Several short/minor comments about the description of the calculations: 29 L22: instead of "particles with diameter D" (should at least be Dp) and since the formation rate 30 31 is not calculated for different particle diameters, I would clearly mention Dp = 9nm, otherwise 32 one has to wait until Section 3.2 to explicitly get this information (and it would also be more 33 consistent with the description of the terms of Eq. (1));

| 1              |                                                                                                                                                                                                                                                                                                         |  |
|----------------|---------------------------------------------------------------------------------------------------------------------------------------------------------------------------------------------------------------------------------------------------------------------------------------------------------|--|
| 2
| -We modified the sentence in Line 22 as "). Formation rates of particles with diameter $D_p$ (in this study $D_p$ =9nm) were calculated"                                                                                                                                                                |  |
| 4              |                                                                                                                                                                                                                                                                                                         |  |
| 5
| L25: "CoagS is the coagulation of particles in this size range" (should be $CoagS_{Dp}$ ): which (lowest) particle size was used to calculate $CoagS_{Dp}$ ? I would suggest a more accurate formulation, such as "CoagS Dp is the coagulation sink of XX nm particles on larger particles"; |  |
| 8              |                                                                                                                                                                                                                                                                                                         |  |
| 9
| -The sentence was modified as "CoagS $_{Dp}$ is the coagulation of 9nm particles on larger particles"                                                                                                                                                                                                   |  |
| 11             |                                                                                                                                                                                                                                                                                                         |  |
| 12             | - L27: Please refer the reader to Dal Maso et al. (2005) for the mode fitting method;                                                                                                                                                                                                                   |  |
| 13             |                                                                                                                                                                                                                                                                                                         |  |
| 14             | A reference to Dal Maso et al., 2005 was added                                                                                                                                                                                                                                                          |  |
| 15             |                                                                                                                                                                                                                                                                                                         |  |
| 16
| L31: For this first occurrence, instead of "the sulfuric acid sink", I would suggest to rather write something more explicit like "CS is the condensation sink caused by the pre-existing aerosol population and was calculated using the characteristics/properties of sulfuric acid".                 |  |
| 19             |                                                                                                                                                                                                                                                                                                         |  |
| 20
| -We modified the sentence as "CS is the condensation sink caused by the pre-existing aerosol population and was calculated using the properties of sulfuric acid as condensing vapor."                                                                                                                  |  |
| 22             |                                                                                                                                                                                                                                                                                                         |  |
| 23
| P6, L11: "relevant chemical reactions": I would recommend to add few words on the relevance of the reactions, at least mention they are related to sulfuric acid production.                                                                                                                            |  |
| 25             |                                                                                                                                                                                                                                                                                                         |  |
| 26
| -We also used reactions for the production of organic compounds except of sulfuric acid so the sentence was changed as "For the present study, chemical reactions relevant to the production of condensing species from the Master Chemical Mechanism"                                                  |  |
| 29             |                                                                                                                                                                                                                                                                                                         |  |
| 30
| P6, L26-27: what is "free form nucleation"? I would suggest to briefly recall the parameterization which is used in the model and introduce the "nucleation coefficient", later                                                                                                                         |  |

discussed in Section 3.5 (P16, L16 & L24-25).

-We introduced the nucleation coefficient and changed the sentence as follow: "UHMA
simulated new cluster formation using the activation nucleation parameterization, so that the
nucleation rate has a linear relationship with sulfuric acid concentration, depending on the
nucleation coefficient Kact."

P6, L29-30: "All these compounds were treated as sulfuric acid and organics": what does this
sentence mean? Also, on L27, if ELVOCs are considered please add "20 extremely lowvolatility organic compounds", otherwise change to "LVOCs."

**10**

-The word "extremely" was added since we actually refer to ELVOCs. All condensing species
were treated either as sulfuric acid if inorganic or organic compounds and this is now made
clear in the text "All condensing compounds were treated either as sulfuric acid or organic compounds and.."

**15**

P7, L6-7: I am a bit confused with this sentence: only the particle size distributions are used to
initialize the model (as reported on P6, L24-25), which then calculates a CS based on the
simulated distributions, right? If the purpose of the abovementioned sentence is only to precise that SMPS data were used to calculate the CS, I would strongly recommend to move it to

Section 2.1 (P5, L31), as Section 2.2 is dedicated to model description.

**21**

- -The sulfuric acid condensation sink is calculated based on measured size distributions and not
   the simulated, this is correctly stated in the text.
- 24

P8, L11-13: I am a bit confused with the use of TUV: was the parameterisation used instead of
 TUV, or implemented in TUV?

**27**

-The parametrization from Mogensen et al., 2015 was used which provides improvement to the
calculation from TUV. We added "...and used in the model." at the end of the sentence.

**30**

P8, L21-27: I am somewhat sceptical about the values which are reported in this paragraph; I
think they do not give much information since the shape of the particle size distribution is highly
variable with respect to seasons, event vs non-event days, time of the day... I would thus
suggest to either provide a more detailed description/comparison of the concentrations in the
different modes and their contribution to total concentration, or at least provide

| 1 | quartiles/standard | deviation | for all | reported | values | (not | only | for | nucleation | mode | particle |
|---|--------------------|-----------|---------|----------|--------|------|------|-----|------------|------|----------|
| 2 | concentration).    |           |         |          |        |      |      |     |            |      |          |

| 3 |                                                                        |
|---|------------------------------------------------------------------------|
| 4 | -We have added standard deviation to all reported mode concentrations. |
| 5 |                                                                        |

P8, L31: are the times local or UTC?

-Thank you for pointing out that the time description is missing. All times are UTC+2 and this
has been added to the captions of the Figures.

P8, L32 - P9, L1: "Such an observation suggests that the nucleation particle number
concentration is controlled by NPF episodes". Isn't it what we expect by definition? Which
other sources would the authors expect for particles in this size range? This comment also refers
to P2 L16-17, P9 L25, P11 L21-22, P15 L22-23. Moreover, concerning the statement P11 L2022, I am not sure if the linear relation between J9 and nucleation mode particle concentration
(N9-25) can be considered as a strong support for NPF being the main source of nucleation
particle, since according to Eq. (1) J9 calculation includes N9-25 in two of the three terms.

-We refer to combustion sources of nucleation mode particles that may play significant role in
polluted areas. At Finokalia we claim that there are no such sources and therefore all
nucleation mode particles observed come from nucleation processes. We have added "rather
than other sources such as local combustion processes" in the text to make it clear. If other
sources than regional NPF contributed significantly that would be evident both in the diurnal

*cycle and the scatter plot of J*9 *and N*nuc.

P9, L10-23: I think that even if deep investigation of night time events is not in the scope of 27 this paper, slightly more detailed description could be provided. In specific: - L10-12: Even if 28 similar night time concentrations are observed during all seasons, they seem to result from 29 different processes based on Fig. 2a. Indeed, there is an increase of the concentrations after 30 18:00 in summer and autumn, which may suggest evening time new particle formation, but 31 during spring and winter the concentrations keep on decreasing until they reach the night time 32 value, suggesting that evening events are not frequent during these seasons. I would thus 33 suggest to balance the sentence from L10-12, and maybe provide frequencies of occurrence of 34 such events for each season, which will also help quantifying "Frequently" (L13).

-We have modified the second sentence of this paragraph "This suggests that there is some new particle production mechanism at night, especially in summer and autumn,...". However, we prefer not to go into further detail as this is work in progress and these events lack the
characteristics of regional NPF that are the focus of this study. There is a description of such events in Kalivitis et al., 2012 that is already cited here.

L17-18: I would also add that on top of the "local" character of these events, which may partly
explain the limited source of condensing vapours (and therefore particle growth), the absence
of photochemistry during night time most likely strengthen the lack of vapours needed to
sustain particle growth.

-This is a very important remark and we appreciate this comment. We added at the end of the
 sentence "and that the lack of photochemistry during night limits the abundance of condensable
 we are driving a particle growth".

*vapors driving particle growth*".

P10, L3-18: I wouldn't say that ozone is "the major oxidant in the atmosphere", especially when
focussing on daytime NPF events, during which OH is expected to play a significant (major?)
role. Also, I don't think that based on the variables included in this factor analysis it is possible
to state that NPF is not sensitive to "atmospheric chemical composition"; compounds other than
ozone such as for e.g. NOx, SO2, monoterpenes... would be needed to draw such conclusions.

-We have rephrased the sentences so that "ozone concentrations (as an important oxidant in
 the atmosphere)" and with regard to the conclusions "...NPF is not sensitive to local meteorological conditions, preexisting particulate matter and ozone levels in this environment.

P11, L3: "the particle survival probability seems to be the highest in winter": the authors have
the data needed to actually test their hypothesis and provide a more robust conclusion, and even
quantify the variations of the survival probability in different seasons.

We calculated the CS/GR ratio for all Class I events and we found it to be smaller in winter
than spring and autumn but surprisingly larger than in summer. This was included in the text.

P11: While they peak at slightly different times of the year, the maximum of the NPF frequency, particle formation and growth rates are all attributed to enhanced biogenic emissions and/or

- 35 photochemistry
- (P10 L27, P11 L16-17 and P11 L25-26, respectively). This hypothesis seems plausible as all maxima are observed during spring/summer, but could the authors comment on the different seasonal variations of the abovementioned variables? In contrast it can be seen from Fig. 1a
 and 6b that the GR and CS have similar seasonal patterns: is it then realistic to think that CS
 and the vapours involved in particle growth "share the same origin"?

-NPF frequency is maximum in mid -spring and early summer. The biogenic activity and the 6 onset of intense photochemistry seem to play a key role in the formation of new particles. 7 During summer however, despite the fact the GR is observed to be the highest for new particles, 8 transported pollutants accumulating in the atmosphere due to the lack of precipitation result 9 to the highest CS, suppressing the formation of new particles. Rain season in southeastern 10 Europe in early autumn leads to gradual CS decrease, and as a result a local maximum in NPF 11 frequency is observed in October. In the revised version of the manuscript that three more years 12 of analysis have been included it was found that the average formation rates have higher values 13 during December, January and March, when the CS is lower. This observation changes the 14 above mentioned general remark that photochemical activity and biogenic emissions are the 15 drivers for the formation rates-the preexisting particle population scavenging precursors is 16 probably defining how fast the new particles form-the lowest formation rates are observed in 17 summer until early autumn. The exact opposite is observed for GR, higher values are observed in summer and September and lowest in winter and March. Photochemistry and biogenic 18 19 emission are probably driving the growth process. However, transported pollution may 20 contribute except of CS to GR as well, transported anthropogenic SO2 may play a role in the 21 growth process as indicated later on when discussing trends. In any case, the minimum values 22 of GR are observed in months that both biogenic and photochemical activity are lowest, and 23 hence condensing vapors are scarce. These information are now included in the text.

25

P11, L27-28: It is true that based on Fig 2a the average duration of NPF in summer seems to be
shorter compared to other seasons, but also the maximum of the concentration is lower. Since
the CS (and consequently CoagS) is also higher during summer (Fig. 1a), I would think that
both the CS (CoagS) and the GR are affecting the variation of nucleation mode particle
concentration (should be checked by calculating the survival probability).

-The survival probability for nucleation mode particles for Class I events was calculated. It 33 was found that on seasonal basis the median survival probability is higher in summer, however 34 varies within 5% and therefore no safe conclusions can be made. On monthly basis the 35 variability was within 13% with higher values observed in November. Nevertheless, we agree that the CS (and hence CoagS) may also affect the maximum concentrations observed. We 36 37 hence modified the sentence as "The average duration of the NPF in summer seems to be 38 shorter and the maximum concentrations of nucleation mode particles during the summer 39 events are lower as shown in Figure 2a. These observations may be explained by the higher 40 GR and CS during summer."

P12, L1, L5: I would slightly balance the statements ("notable increase", "clear decreasing") as
in my opinion the reported observations are not as obvious as suggested.

We have modified the whole paragraph since we included additional years in our analysis. In
any case, we use modest expressions for our statements regarding the trends.

P12, L5-33: I am not fully convinced by the conclusion reported on L30, which suggests that
 decreased SO2 concentrations related to the economic crisis in Europe may explain observed
 variations of GR and occurrence of class I NPF events. Main reasons for this are listed below:

The lack of SO2 measurement in Finokalia prevents from any direct evaluation of the SO2 52 concentration decrease at this site;

Based on previous studies mentioned in the present work it seems that decreasing SO2

concentrations can lead to contrasting observations, thus pointing to the fact that robust conclusions cannot be inferred from the analysis of SO2 alone;

While the important role of H2SO4 in early nucleation stage has been reported in different studies, the need for other species to explain observed GR has also been evidenced, and the present paper itself tends to emphasize the role of organic species in NPF at Finokalia. Indeed, 4 maximum of NPF occurrence, J9 and GR are all attributed to enhanced biogenic emissions, and 5 best agreement between model simulation and observation is achieved when adjusting 6 monoterpenes concentration in the model. I would thus think that SO2 driving the observed 7 variations of GR and NPF occurrence is not fully consistent with the aforementioned 8 observations/results.

-Given the objections of the reviewer, in the revised version of the manuscript we have
 rephrased the sentence, so that it simply provides to the reader the information that since the
 outbreak of the economic crisis we have observed changes in the atmospheric composition that
 could influence the vapors involved in NPF processes.

P13, L12-14: Does this sentence mean that instrument malfunction was affecting measurement of positive ions? If not, it is fine to focus on negative ions only, but I wouldn't justify this choice based on their better ability to represent NPF events. Indeed, it is in my opinion complex to assess which polarity gives the "better representation of NPF events", as the different observations from the two DMAs may instead reflect the signature of the nucleation mechanism.

No, it does not indicate malfunction of the AIS instrument. The observation of NPF was more
evident in the negative polarity and this has been reported in earlier work (Kalivitis et al.,
2012) that was cited.

P13, L17-26: I would have expected the AIS-derived NPF frequencies to be more often higher 27 (or at least equal) than SMPS-derived ones, while the opposite is shown on Fig. 10. Does it 28 mean that the event day illustrated on Fig. 9 is only representative of a rather limited fraction 29 of the events observed in Finokalia, while the majority of them is actually not visible from the AIS smallest diameters? In order to make the most of the FRONT dataset and provide more 30 31 information on the nature of the events detected in Finokalia during this period, I would suggest 32 to also report for each month (on Fig. 10 for instance) the number of events detected by each 33 instrument and the number of event days they have in common. This will help assessing the 34 fraction of events with very limited growth only visible in AIS data, the fraction of regional 35 events detected by both instruments and that of events only visible in SMPS data..

-We have modified the Figure 12 so that the event days are mentioned on top of each month
that present NPF for AIS, SMPS and the common days. Indeed the NPF events are less in AIS
than SMPS .This has been reported in the K-puszta station in Hungary (Yli-Juuti et al., 2009),
probably because AIS detects only naturally charged particles while SMPS all particles. The
reference was introduced in the manuscript.

P14, L6-33: I have several comments/questions regarding model simulations:

L15: What does "NPF level" mean?46

-This was wrong expression , we replaced it with "NPF events".

L22-26: Could the authors briefly summarize the strategy they adopted to finally reach fair
agreement between model simulation and observation? For instance, which sensitivity tests
were performed, were parameters other than nucleation coefficient and monoterpenes
concentration also tuned?

COIR

-The approach was quite simplistic: to adjust the nucleation coefficient and the monoterpene
 concentrations so that we simulate efficiently the nucleation and growth rate observed during the second day of the "event week" when the most pronounced NPF event was observed. This 2 is now also described in the manuscript. 3 L25: are the levels of final simulated monoterpenes concentration realistic, are they for instance 4 5 in agreement with observations from 2014? 6 7 -Yes, the values are realistic and they compare well with the findings of Debevec et al., 2018 8 that measured monoterpenes during NPF events in eastern Mediterranean (Cyprus). This is 9 now stated in the text. 10 11 L27-29: I would slightly balance the conclusions ("well captured", "in such detail"), as if I 12 agree with the fact that the reported results are very encouraging, one can observe some 13 discrepancies between model and observation (e.g. NPF event from day 243 in not visible in 14 model data); 15 -We have tried to balance the conclusions by removing these expressions. 16 L29-31: Do the authors also consider the possibility to test other nucleation mechanisms in 17 18 future simulations? 19 20 -Yes, we plan to continue simulating NPF at Finokalia and introduce actual VOC 21 measurements within 2019. We added at the last sentence ", new simulations and VOC 22 measurements will further provide insight in the nucleation mechanisms, the growth process 23 and the factors controlling NPF in the eastern Mediterranean atmosphere." 24 25 References: Mirme, A., E. Tamm, G. Mordas, M. Vana, J. Uin, S. Mirme, T. Bernotas, L. Laakso, A. 26 Hirsikko, and M. Kulmala: A wide-range multi-channel Air Ion Spectrometer, Boreal Environ. 27 28 Res., 12, 247–264, 2007.

Vanhanen, J., Mikkilä, J., Lehtipalo, K., Sipilä, M., Manninen, H. E., Siivola, E., Petäjä, T.,
and Kulmala, M.: Particle size magnifier for Nano-CN detection, Aerosol Sci. Tech., 45, 533–
542, 2011.

-References

-Debevec, C., Sauvage, S., Gros, V., Sellegri, K., Sciare, J., Pikridas, M., Stavroulas, I.,

Leonardis, T., Gaudion, V., Depelchin, L., Fronval, I., Sarda-Esteve, R., Baisnée, D., Bonsang,

B., Savvides, C., Vrekoussis, M., and Locoge, N.: Driving parameters of biogenic volatile organic compounds and consequences on new particle formation observed at an eastern 4 Mediterranean background site, Atmos. Chem. Phys., 18, 14297-14325,

https://doi.org/10.5194/acp-18-14297-2018, 2018. 6

-Yli-Juuti, T., Riipinen, I., Aalto, P. P., Nieminen, T., Maenhaut, W., Janssens, I. A., Claeys,

M., Salma, I., Ocskay, R., Hoffer, A., Imre, K. and Kulmala, M.: Characteristics of new particle formation events and cluster ions at K-puszta, Hungary. Boreal Env. Res. 14: 683-9 10 698, 2009.

**1 Response to Reviewer #2**

**1 - Response to reviewer #2**

The authors would like to thank the reviewer for the comments that helped to improve this 3 manuscript. Please find below a point-by-point reply to all of the issues raised and the 4 corresponding changes.

Formation and growth of atmospheric nanoparticles in the eastern Mediterranean: Results
from long-term measurements and process simulations" reports long term data from a Station
in South Europe. Whilst the data are of good quality and worth publication (long term smps
data are scarce) the analysis is fair and does not add any new result. The increasing trend (and
decreasing trend) or NPF and GR (respectively) may suffer from lack of data at the beginning
of the period (2008-2012) relative to the last part of the period (2012-2015) - the trend may be
a simple artifact.

-In order to address this issue we chose to expand our analysis until 2018 in order to have ten
years of data. As you can see in the Picture 1 at the end of this response, indeed there were less
data available at the beginning of period under study. Nevertheless, there was always at least

- 17 70% coverage of each year and total coverage 82%.
- 18
- 19 line 8 abstract : biogenic marine or land or both? specify
- 20
- 21 -The terrestrial biogenic activity is expected to contribute more efficiently to NPF and this is
now stated in the text.
- 23
- line 11-13 Do not understand what the sentence means. please reprhase and specify simply yousee NPF during night time (seen elasewhere too).
- 26
- 27 -The sentence was changed to "Throughout the period under study, nucleation was observed
also during the night."
- 29
- 30 sentence 18-22 not very clear maybe concomitant ion size distributions suggests
- 31
- We have rephrased the sentence to "Classification of NPF events based on ion spectrometer
   measurements differed from the corresponding classification based on a mobility
   spectrometer,.."

pg 3 perhaps report the study of Dall Osto et al (2018, Sci. Rep.) reported by same co-authors
showing south Europe is different from Central and North Europe.

-We have added the sentence" It has been shown that the processes responsible for particle
formation and growth differ substantially across the European continent (Dall' Osto et al.,
2018)."

pg 8 line 16-20 I think this is not correct, likely the longest SMPS size distrubutions are likely10 in Barcelona and regional areas of Montsein (Dr. Querol s group).

-We changed the sentence to "one of the longest time series".

Increase NPF events and decreased GR - this is interesting, it makes sense if the CS is lower
 over time, there is likely more NPF events, and they likely grow less cause likely you have less
 condensing material.

-As explained at the following comment this trend is not observed in the updated ten year
 analysis. However, we consider the observation for the period 2008-2015 important given the
 measurements availability.

Figure 8a. I think the whole conclusion may simply be noise. If looking at figure 8a, you see
2008-2010 you have less datapoints (perhaps in spring - summer) that causes the trend you may
have. It looks if you remove the 2008-2009, the trend to me is not existing. I would be careful
to say there is a trend (and so I would remove and change all abstract) - it is visually clear that
years 2008-2010 have smaller data coverage that 2013-2015.

-The trends actually disappeared while including the additional years so that the time series covers form 2008 to 2018. If we remove the first two years as suggested, an opposite trend is revealed that it statistically significant and it is described in the manuscript, a clear decreasing 31 trend in the period 2010-2018. The additional years added in the analysis showed that 1) we trend in the period 2010-2018. The additional years added in the analysis showed that 1) we 32 had a period of increased NPF frequency in 2010-2014, 2) there is a decreasing trend since until today 3) the decreasing trend of GR did not continue, however for the period 2008 -

2015 it was statistically significant. These are all now stated in the manuscript. Since however they cannot be expanded for the whole timeseries they are removed from the abstract and the
 concluding marks as recommended.

Considering the above, I see this study does not add much additional novel results, although it is worth publication cause you clearly see a long SMPS time trend showing spring nucleations (different from typical summer ones).

 a
 Month
 b
 Ioan

 Picture 1: Size distribution data availability at Finokalia, Greece during the period June 2008-June2018 on monthly

 basis (a) and interannualy (b)

**1 List of Changes made**

- 1 The abstract was changed according to suggestions
- 2 All data presented have been expended for the period 2008-2018, so that data for
- 3 the period 2015-2018 was added
- 4 Changes in paragraphs 3.2 and 3.3 were made according to suggestions and new
- 5 findings
- 6 Tables were changed
- 7 All figures except of 2 and 12 were updated or changed.
- 8 Several minor edits across the manuscript
- 9
- 10
- 10

**1 Marked up manuscript version**

| 1        |                                                                                                                                                     |
|----------|-----------------------------------------------------------------------------------------------------------------------------------------------------|

[revised manuscript text omitted]
_{D\rho} = \frac{\Delta N_{Dp}}{\Delta t} + CoagS_{Dp} \cdot N_{Dp} + \frac{GR}{\Delta D_p} \cdot N_{Dp} + S_{losses} (1)$$

 $\Delta N_{D_p}$  is the increase in nucleation mode particles' number concentration (Dp<25nm), CoagSDp is the coagulation of 9nm particles on larger particles, GR is the growth rate in the size range 9-25nm. Slosses takes into account additional losses and was neglected in this study. GR was calculated using the mode-fitting method, (Dal Maso et al., 2005). The aerosol size distributions were fitted with lognormal distributions and the nucleation mode geometric mean diameter was plotted as a function of time. GR was calculated as the slope of the linear fit so that: Formatted: Hyperlink, English (United States)

| Deleted: CoagS                     |
|------------------------------------|
|                                    |
| Deleted: in this size range |
|                                    |
|                                    |
| Deleted:                           |

 $GR = \frac{dD_p}{dt} (2)$

CS is the condensation sink caused by the pre-existing aerosol population and was calculated using the properties of sulfuric acid as condensing vapor.

All important meteorological parameters were monitored every five minutes using an automated meteorological station, including the temperature, wind velocity and direction, relative humidity, solar irradiance and precipitation. Ozone concentrations were measured with a TEI 49C instrument and nitrogen oxides with a TEI 42CTL, both commercially available, with a time step of five minutes.

2.2) NPF simulations with the MALTE-Box model

The simulations of NPF events in the eastern Mediterranean atmosphere were here 11 performed using the MALTE-box model of the University of Helsinki. This 0-d model able to 12 simulate aerosol dynamics and chemical processes has successfully reproduced observations 13 of aerosol formation and growth in the boreal environment (Boy et al., 2006) as well as in 14 highly polluted areas (Huang et al., 2016). For the present study, chemical reactions relevant to the production of condensing species from the Master Chemical Mechanism were 15 incorporated in the MALTE-box chemical mechanism, as described in Boy et al. (2013). These 16 17 include the full MCM degradation scheme of the following volatitle organic compounds 18 (described in more detail in Tzitzikalaki et al., 2017): C1-C4 alkanes, C2-C3 alkenes, acetylene, isoprene,  $\alpha$ - and  $\beta$ -pinene, aromatics, methanol, dimethyl sulfide, formaldehyde, formic and 19 20 acetic acids, acetaldehyde, glycoaldehyde, glyoxal, methylglyoxal, acetone, hydroxyacetone, 21 butanone and marine amines. The Kinetic PreProcessor (KPP) was used to produce the Fortran 22 code for the calculations of the concentrations of each individual compound (Damian et al., 2002), except for those species whose concentrations were manually input from large scale 23 model simulations. 24

The major aerosol dynamical processes for clear sky atmosphere were simulated by the sizesegregated aerosol model UHMA (University Helsinki Multicomponent Aerosol Model, Korhonen et al., 2004) impended in the MALTE-Box model. Measured aerosol number size distributions were used to initialize UHMA daily, which simulates NPF, coagulation, growth and dry deposition of particles. UHMA simulated new cluster formation using the activation nucleation, parameterization, so that the nucleation rate has a linear relationship with sulfuric

 $\label{eq:acid} \textbf{31} \qquad \text{acid concentration, depending on the nucleation coefficient } K_{\text{act.}}$

| Deleted: sulfuric acid             |
|------------------------------------|
| Deleted: preexisting               |
| Deleted: with unit s -1 |

Apart from sulfuric acid, about 20 extremely low-volatility organic compounds (ELVOCs) and

7 selected semi-volatile organic compounds (SVOCs) were treated as condensing vapors, following the simplified chemical mechanism presented in Huang et al. (2016). All condensing- compounds were treated either as sulfuric acid or organic compounds and the condensation of organic vapors was determined by the nano-Kohler theory (Kulmala et al., 2004b).

As input to the MALTE-Box model were used the observations at Finokalia station and when 7 such observations were not available, the results from numerical simulations with the global 8 3-dimensional chemistry transport model (CTM) TM4-ECPL (Daskalakis et al., 2015, 2016; 9 Myriokefalitakis et al., 2010, 2016) for Finokalia. Observational data include temperature, 10 relative humidity, total radiation (meteorological input), ozone ( $O_3$ ) and nitrogen oxides (NOx) 11 concentrations as well as aerosol number size distributions. The aerosol number size 12 distribution measured by the SMPS was used to calculate the condensation sink for H2SO4 vapors. Due to the lack of detailed measurements of VOC at Finokalia, as a first approximation, 13 biogenic and anthropogenic concentrations of all the above mentioned VOCs resolved every 14 3 hours were taken from the TM4-ECPL model. 15

The global TM4-ECPL model was run driven for this study by ECMWF (European Centre for 16 17 Medium – Range Weather Forecasts) Interim re-analysis project (ERA – Interim) meteorology (Dee et al., 2011) of the year 2012 at an horizontal resolution of 3° in longitude x 2° in latitude 18 19 with 34 vertical layers up to 0.1 hPa. The model used year-specific meteorology and emissions 20 of trace gases and aerosols. For this study, that of the year 2012 was used, except for soil NOx 21 and oceanic CO and VOCs emissions which were taken from POET inventory database for the 22 year 2000 (Granier et al., 2005). TM4-ECPL simulations for this work were performed with a 23 model time-step of 30 min, and the simulated VOC concentrations every 3-hours were used 24 as input to MALTE box model; while SO2 surface levels at Finokalia were taken from Monitoring 25 Atmospheric Composition and Climate (MACC) data assimilation system (Inness et al., 2013).

For the calculations of the photo-dissociation rate coefficient by the MALTE-Box model, the solar actinic flux (AF) is needed. Unfortunately, AF was not measured at Finokalia in 2012, therefore AF levels were calculated by the Tropospheric Ultraviolet and visible Radiation Model (TUV, Madronich, 1993) version v.5 for cloud free conditions. The ability of TUV to calculate the AF at Finokalia was investigated by comparing observations of photo dissociation rates of O3 (JO1D) and NO2 (JNO2) and model calculations. The measurements of these photo dissociation rates were performed by filter radiometers (Meteorologie Consult, Germany).

| - | Deleted: vapours      |
|---|-----------------------|
| - | Deleted: these        |
| - | Deleted: and organics |

The JO1D was measured at wavelengths <325nm, while for JNO2 wavelengths <420nm were</li>

[revised manuscript text omitted]
                                                                                                          |
|   | Deleted: , even though                                                                                                                                                                           |
|   | Deleted: very                                                                                                                                                                                    |
|   | Deleted: 1±                                                                                                                                                                                      |
|   | Deleted: (median 4.1 nm hr -1 ).                                                                                                                                                      |
|   | Formatted: Font color: Text 1                                                                                                                                                                    |
|   | Deleted: The average duration of the NPF                                                                                                                                                         |
|   | Formatted: Font color: Text 1                                                                                                                                                                    |
|   | Deleted: seems                                                                                                                                                                                   |
|   | Formatted: Font color: Text 1                                                                                                                                                                    |
|   | Deleted: be shorter                                                                                                                                                                              |
|   | Formatted: Font color: Text 1                                                                                                                                                                    |
|   | Deleted: shown                                                                                                                                                                                   |
|   | Formatted: Font color: Text 1                                                                                                                                                                    |
|   | Deleted: Figure 2a and that may be explained by the higher                                                                                                                                |
|   | Formatted: Font color: Text 1                                                                                                                                                                    |
|   | Deleted: rates observed.                                                                                                                                                                         |

The survival probability of newly-formed particles is closely related to the ratio of CS to GR, at 2 least for cluster sizes (Kerminen and Kulmala, 2002; Kulmala et al., 2017) and at Finokalia they present the same annual cycle. The survival probability for nucleation mode particles for Class 3 I events was calculated based on the formula in Kulmala et al. (2017). It was found that on 4 5 seasonal basis the median survival probability is higher in summer and winter, however varies 6 between the seasons only within 5%. The concentrations of nucleation mode particles are 7 lower during summer and the average duration of the NPF in summer seems to be shorter as shown in Figures 1 and 2a respectively. These observations may be explained by the higher CS 8 9 and GR during summer. The CS (and hence CoagS) may directly affect the maximum 10 concentrations observed. The slightly higher survival probability in summer explains perhaps 11 that given the high CS, in order for new particles to survive the need to grow fast. On the other 12 hand, one would expect NPF to be most frequent in winter when the highest concentrations 13 of nucleation particles are observed and CS is the lowest, however this was not the case. A 14 possible explanation for the high nucleation mode particle number concentrations in winter 15 could be that the survival probability is higher than in spring or autumn.

**16**

**17 3.3) NPF trends during the 2008-2018 period**

During the period under study no statistically significant trends in NPF events were observed 18 19 at Finokalia for the 120 available months. It should be noted though, that since 2010 a 20 decreasing trend is observed, which is statistically significant with a p-value of 0.005. During 21 the measurement period under study, no trend in  $J_9$  was observed (Figure 8c). Although no 22 statistically significant trend was observed for GR9-25 as well (Figure 8d), we observed a 23 decreasing trend during the period 2008-2015 of about 0.3 nm  $hr^{-1}yr^{-1}$ . This trend can be considered statistically significant (p-value of 0.03). In order to explain this trend, we need to 24 25 emphasize the regional characteristics of the observations at Finokalia, as this site is greatly 26 affected by long-range transported pollutants of marine, desert dust and polluted continental 27 origin (Lelieveld et al., 2002). Non-sea salt sulfate (nss-SO42-) can be considered as an indicator 28 of regional pollution from anthropogenic activities (SO2 emissions), and since the beginning of 29 the economic crisis in Europe, especially in Greece, we observed a clear decline in its 30 concentration since 2008 (Paraskevopoulou et al., 2015) which however has stopped after 2015. We can therefore assume also a regional decrease in SO2 emissions, since the main 31 32 source of SO2 at Finokalia is attributed to transported pollution (Sciare et al., 2003). This could

**Deleted: 2015**

| Deleted: By looking at the inter-annual evolution of the NF monthly event frequency at Finokalia for the 85 available months, we observe a slight increase of about 1.5 % per year (Figure 8a). This trend is not |  |  |  |  |  |
|--------------------------------------------------------------------------------------------------------------------------------------------------------------------------------------------------------------------------|--|--|--|--|--|
| Formatted: Font color: Auto                                                                                                                                                                                              |  |  |  |  |  |
| Formatted: Font color: Auto                                                                                                                                                                                              |  |  |  |  |  |
| Deleted: 07. This increase is a result of a notable increase o
Class II NPF events despite a simultaneous decrease of Class
events from 2008 to 2015 (8b).                                                  |  |  |  |  |  |
| Formatted: Font color: Auto                                                                                                                                                                                              |  |  |  |  |  |
| Deleted: was found to be                                                                                                                                                                                                 |  |  |  |  |  |
| Formatted: Font color: Auto                                                                                                                                                                                              |  |  |  |  |  |
| Formatted: Font: Not Bold, Font color: Auto                                                                                                                                                                              |  |  |  |  |  |
| Formatted: Font color: Auto                                                                                                                                                                                              |  |  |  |  |  |
| Deleted: , but the winters 2008-9 and 2012-13 had clearly higher values of $J_9$ than the rest of the time                                                                                                               |  |  |  |  |  |
| Formatted: Font color: Auto                                                                                                                                                                                              |  |  |  |  |  |
| Deleted: ¶
When looking at the temporal variation of GR                                                                                                                                                               |  |  |  |  |  |
| Formatted: Font color: Auto                                                                                                                                                                                              |  |  |  |  |  |
| Deleted: observe                                                                                                                                                                                                         |  |  |  |  |  |
| Formatted: Font color: Auto                                                                                                                                                                                              |  |  |  |  |  |
| Deleted: clear                                                                                                                                                                                                           |  |  |  |  |  |
| Formatted: Font color: Auto                                                                                                                                                                                              |  |  |  |  |  |
| Formatted: Font color: Auto                                                                                                                                                                                              |  |  |  |  |  |
| Deleted: , the no-trend hypothesis test returned a                                                                                                                                                                       |  |  |  |  |  |
| Formatted: Font color: Auto                                                                                                                                                                                              |  |  |  |  |  |
| Deleted: .                                                                                                                                                                                                               |  |  |  |  |  |
| Formatted: Font color: Auto                                                                                                                                                                                              |  |  |  |  |  |
| Deleted: can observe                                                                                                                                                                                                     |  |  |  |  |  |
| Formatted: Font color: Auto                                                                                                                                                                                              |  |  |  |  |  |
| Formatted: Font color: Auto                                                                                                                                                                                              |  |  |  |  |  |
| Deleted: ).                                                                                                                                                                                                              |  |  |  |  |  |
| Formatted: Font color: Auto                                                                                                                                                                                              |  |  |  |  |  |
| Deleted: a major part                                                                                                                                                                                                    |  |  |  |  |  |
| Formatted: Font color: Auto                                                                                                                                                                                              |  |  |  |  |  |
| Deleted: can be                                                                                                                                                                                                          |  |  |  |  |  |
| Formatted: Font color: Auto                                                                                                                                                                                              |  |  |  |  |  |
| Deleted: would                                                                                                                                                                                                           |  |  |  |  |  |
| Formatted: Font color: Auto                                                                                                                                                                                              |  |  |  |  |  |

[revised manuscript text omitted]

for NPF events observed at Finokalia and condensational sink for sulfuric acid (CS) on seasonal base during the period June 2008-June 2018 (mean, median and standard deviation).

- 10
- 11

|   | Deleted: 623          |                                         |
|---|-----------------------|-----------------------------------------|
|   | Deleted: 29.37        |                                         |
|   | Deleted: 161          |                                         |
|   | Deleted: 59           |                                         |
|   | Deleted: 462          |                                         |
|   | Deleted: 21.78 |                                         |
|   | Deleted: 555          |                                         |
|   | Formatted             |                                         |
|   | Deleted: 26.17        |                                         |
|   | Deleted: 943          |                                         |
|   | Deleted: 44.46        |                                         |
|   | Deleted: 2121         |                                         |
|   | Deleted: 2015         |                                         |
|   | Formatted             |                                         |
|   | Deleted: 7            |                                         |
|   | Formatted             |                                         |
| 4 | Formatted             |                                         |
| / | Deleted: 0            |                                         |
|   | Formatted             |                                         |
|   | Deleted: 7            | _                                |
|   | Deleted: 2            | ~~~ |
|   | Formatted             |                                         |
|   | Deleted: .7           |                                         |
|   | Formatted             |                                         |
|   | Deleted: 1            |                                         |
|   | Deleted: 6            |                                         |
|   | Formatted             |                                         |
|   | Deleted: .9           |                                         |
|   | Formatted             |                                         |
|   | Deleted: 8            |                                         |
|   | Formatted             |                                         |
|   | Deleted: .6           |                                         |
|   | Formatted             |                                         |
|   | Deleted: 3.5          |                                         |
|   | Formatted             |                                         |
|   | Deleted: 2.7          |                                         |
|   | Formatted             |                                         |
|   | Deleted: 2.5          |                                         |
|   | Formatted             |                                         |
|   | Deleted: 5            |                                         |
|   | Formatted             |                                         |
|   | Deleted: 1            |                                         |
|   | Deleted: 2.7          |                                         |
|   | Formatted             |                                         |
|   | Deleted: 6            |                                         |
|   | Formatted             |                                         |
|   | Formatted             |                                         |
|   |                       |                                         |
|   | Formatted             |                                         |
|   |                       |                                         |
| 1 | Formatted             | [ ]                                     |

---

## Editor Decision (ED1)

**Editor's comments on the revised version of ms. acp-2018-229 by N. Kalivitis et al., entitled "Formation and growth of atmospheric nanoparticles in the eastern Mediterranean: Results from long-term measurements and process simulations"**

François Dulac, 24 January 2019

I thank you for your worthwhile revision including an extension of your dataset. I am pleased to accept your revised manuscript for publication in the ChArMEx special issue in ACP pending a few technical corrections listed hereafter (changes in quoted text are underlined):

- Throughout the paper, use the italic style for all variables (e.g. $J$, $CS$, $CoagS$, $Dp$...)

- Page 2, line 18: "we use the MALTE-box model for simulating a case study".

- Page 2, lines 21-22: "The adjusted parameterization resulting from our sensitivity tests was significantly different from the initial one that had been determined for the boreal environment".

- P.4, l.3: "representing one of the longest".

- P. 5, lines 24 and 26: consider changing $CoagS_{Dp}$ to $CoagS$, used later.

- P. 5, l.26: "of 9-nm particles".

- P.5, l.27: insert a space between number and unit.

- P.6, l.17: specify the meaning of "MCM".

- P.7, l.24: remove "while".

- P.8, l.9: "form the Ozone Monitoring".

- P.8, l.12: it is needed to specify whether downloaded AERONET data are from Version 2 or Version 3 of the product.

- P.8, l.14: insert a space between number and unit (2 occurrences).

- P.11, l.11: do you mean "with a good confidence or not, respectively."?

- P.11, lines 30-31: please consider rephrasing this sentence, presently unclear.

- P.12, l.5: "a seasonal basis".

- P.12, l.10-11: "perhaps explains that given high $CS$ values, new particles need to grow fast in order to survive".

- P.14, lines 5-6: check the sentence, you may have to remove "and is probably due and".

- P.14, l.7: "SMPS detects all particles".

- P.14, l.10: insert a space between number and unit.

- P.15, l.23: "also at another location".

- P. 16, l. 7: "in the eastern Mediterranean".

- Legend of tables: Use "Table N. " rather than "Table N) ".

- Legend of Tab. 2: "9-nm particles"; "on a seasonal basis".

- Legend of figures: use "Figure M." rather than "M)"; labels of possible panels must be included with brackets around letters (e.g. "Figure 2. (a) Average […] June 2018. (b) New particle […]").

- Legend of Figs. 4 and 6: specify "the horizontal line in the box".

- Legend of Fig. 7: "9-nm particles"; "for events when $J_9$"; express the ordinate ($N_{9-25}$) as a function of the abscissae ($J_9$) rather than the opposite; you can probably limit the x-axis scale at 10, and limit the y-axis scale at $10^4$ and/or expand the vertical dimension of the plot for a better readability.

- Legend of Fig. 8: "9-nm particles"; I guess that dotted lines in (c) and (d) show the linear regression, specify and check the line which is hardly visible in (d).

- Legend of Fig. 10: "the number of NPF events"; the number of common events"; for better readability of the light grey, rather use black for all numbers and "(top)" for AIs, "(middle)" for SMPS, and "(bottom, italic)" for both instruments; check italic style of the 3 for the month 11-2013.

- Legend of Fig. 11: specify the "event week (i.e. with the most pronounced NPF event observed)".

- Legend of Figs. 11 and 12: "d)" should read "(a)"; rather write "MALTE-box simulations with the adjusted parameters";

- Please apply the Copernicus instructions for figure citations as available at https://www.atmospheric-chemistry-and-physics.net/for_authors/manuscript_preparation.html: The abbreviation "Fig." should be used when it appears in running text and should be followed by a number unless it comes at the beginning of a sentence, e.g.: "The results are depicted in Fig. 5"; Results show […] (Fig. 3); "Figure 9 reveals that".

- Colours: in the web page just cited, Copernicus also recommends keeping colour blindness in mind and avoiding the parallel usage of green and red in maps and charts. For a list of colour scales that are illegible to a significant number of readers, please visit http://colorbrewer2.org/#type=sequential&scheme=BuGn&n=3, and consider changing the colour codes in Figs 2, 3, 7 and 9.

---

## Author Response (AR2)

*Response to Editor's comments.*

-I thank you for your worthwhile revision including an extension of your dataset. I am pleased to accept your revised manuscript for publication in the ChArMEx special issue in ACP pending a few technical corrections listed hereafter (changes in quoted text are underlined):

*The authors would like to thank the Editor and the Reviewers for the useful comments throughout the review process that helped to improve the manuscript in order to be published in the ChArMEx special issue in ACP. Please find below a point-by-point reply (in italic) to all of the technical corrections requested.*

- Throughout the paper, use the italic style for all variables (e.g. $J$, $CS$, $CoagS$, $Dp$…)
*All variables are now displayed in italic style.*

- Page 2, line 18: "we use the MALTE-box model for simulating a case study".
*Text changed according to suggested correction.*

- Page 2, lines 21-22: "The adjusted parameterization resulting from our sensitivity tests was significantly different from the initial one that had been determined for the boreal environment".
*Change made.*

- P.4, l.3: "representing one of the longest".
*Text changed according to suggestion*

- P. 5, lines 24 and 26: consider changing $CoagSDp$ to $CoagS$, used later.
*CoagS is now used throughout the text*

- P. 5, l.26: "of 9-nm particles".
 *Change made.*

- P.5, l.27: insert a space between number and unit.
*A space was added between each number and its units.*

- P.6, l.17: specify the meaning of "MCM".
*The abbreviation MCM is explained earlier and  the abbreviation in parenthesis was added after "Master Chemical Mechanism (MCM)".*

- P.7, l.24: remove "while".
*Done*

- P.8, l.9: "form the Ozone Monitoring".
*Change made.*

- P.8, l.12: it is needed to specify whether downloaded AERONET data are from Version 2 or Version 3 of the product.
*The version was added "Data level 1.5 from Version 2 were used (cloud-screened)."*

- P.8, l.14: insert a space between number and unit (2 occurrences).
*A space was added between each number and its units.*

- P.11, l.11: do you mean "with a good confidence or not, respectively."?
*Yes, that is what was meant and it is now corrected.*
.
- P.11, lines 30-31: please consider rephrasing this sentence, presently unclear.
*The sentence was changed to "Additionally, transported pollution in summer at Finokalia may contribute except of CS to GR as well, since transported anthropogenic SO2 is a precursor for condensable sulfuric acid."*

- P.12, l.5: "a seasonal basis".
*Change made.*

- P.12, l.10-11: "perhaps explains that given high *CS* values, new particles need to grow fast in order to survive".
*The sentence was modified according to suggestion/*

- P.14, lines 5-6: check the sentence, you may have to remove "and is probably due and".
*Indeed, it was removed.*

- P.14, l.7: "SMPS detects all particles".
*Correction made.*

- P.14, l.10: insert a space between number and unit.
*A space was added between each number and its units.*

- P.15, l.23: "also at another location".
*Change made.*

- P. 16, l. 7: "in the eastern Mediterranean".
*Change made.*

- Legend of tables: Use "Table N. " rather than "Table N) ".
*Change made.*

- Legend of Tab. 2: "9-nm particles"; "on a seasonal basis".
*Corrections made.*

- Legend of figures: use "Figure M." rather than "M)"; labels of possible panels must be included with brackets around letters (e.g. "Figure 2. (a) Average […] June 2018. (b) New particle […]").
*Corrections made.*

- Legend of Figs. 4 and 6: specify "the horizontal line in the box".
*Change made.*

- Legend of Fig. 7: "9-nm particles"; "for events when $J9$"; express the ordinate ($N9$-$25$) as a function of the abscissae ($J9$) rather than the opposite; you can probably limit the x-axis scale at 10, and limit the y-axis scale at 104 and/or expand the vertical dimension of the plot for a better readability.

*Both the figure and the legend were modified as suggested.*

- Legend of Fig. 8: "9-nm particles"; I guess that dotted lines in (c) and (d) show the linear regression, specify and check the line which is hardly visible in (d).
*Indeed the line is the regression line, it was highlighted and described in the legend*

- Legend of Fig. 10: "the number of NPF events"; the number of common events"; for better readability of the light grey, rather use black for all numbers and "(top)" for AIs, "(middle)" for SMPS, and "(bottom, italic)" for both instruments; check italic style of the 3 for the month 11-2013.
*The legend was changed according to suggestions, as well as the figure, the common NPF days are now displayed in bold and italic style.*

- Legend of Fig. 11: specify the "event week (i.e. with the most pronounced NPF event observed)".
*In the legend it is now specified that "the "event week" that NPF events were observed at Finokalia."*

- Legend of Figs. 11 and 12: "d)" should read "(a)"; rather write "MALTE-box simulations with the adjusted parameters";
*Corrected in both figures.*

- Please apply the Copernicus instructions for figure citations as available at https://www.atmospheric-chemistry-and-physics.net/for_authors/manuscript_preparation.html: The abbreviation "Fig." should be used when it appears in running text and should be followed by a number unless it comes at the beginning of a sentence, e.g.: "The results are depicted in Fig. 5"; Results show […] (Fig. 3); "Figure 9 reveals that".
*The instructions are now followed throughout the manuscript*

- Colours: in the web page just cited, Copernicus also recommends keeping colour blindness in mind and avoiding the parallel usage of green and red in maps and charts. For a list of colour scales that are illegible to a significant number of readers, please visit http://colorbrewer2.org/#type=sequential&scheme=BuGn&n=3, and consider changing the colour codes in Figs 2, 3, 7 and 9.
*The figures 2.a and 7 were modified so that green and red are not displayed in parallel. However for the contour plots we would prefer to keep the initial color scale, as it is used extensively in the literature for displaying NPF events, and furthermore, all available palettes that were tested provided poor results that did not display the features of NPF as clear as the used scale.*
* * *
*Changes made in the manuscript*

-The changes in the text suggested by the editor were made.

-The legends of the figures were modified as suggested.

-Figures 2.a, 7, 8, 10 were slightly modified (in 2a and 7 the colors were modified, in 8 the regression lines were emphasized, in 10 the number of events font was changed)

*Marked-up manuscript version*

[revised manuscript text omitted]